# Natural variation in *OsMYB8* confers diurnal floret opening time divergence between *indica* and *japonica* subspecies

Yajun Gou [1,4], Yueqin Heng [1,4], Wenyan Ding[1], Canhong Xu[1], Qiushuang Tan[1], Yajing Li[1], Yudong Fang[1], Xiaoqing Li[1], Degui Zhou[2], Xinyu Zhu[1], Mingyue Zhang [1], Rongjian Ye[3], Haiyang Wang [1]✉ & Rongxin Shen [1]✉

The inter-subspecific *indica-japonica* hybrid rice confer potential higher yield than the widely used *indica-indica* intra-subspecific hybrid rice. Nevertheless, the utilization of this strong heterosis is currently hindered by asynchronous diurnal floret opening time (DFOT) of *indica* and *japonica* parental lines. Here, we identify *OsMYB8* as a key regulator of rice DFOT. OsMYB8 induces the transcription of JA-Ile synthetase *OsJAR1*, thereby regulating the expression of genes related to cell osmolality and cell wall remodeling in lodicules to promote floret opening. Natural variations of *OsMYB8* promoter contribute to its differential expression, thus differential transcription of *OsJAR1* and accumulation of JA-Ile in lodicules of *indica* and *japonica* subspecies. Furthermore, introgression of the *indica* haplotype of *OsMYB8* into *japonica* effectively promotes DFOT in *japonica*. Our findings reveal an *OsMYB8-OsJAR1* module that regulates differential DFOT in *indica* and *japonica*, and provide a strategy for breeding early DFOT *japonica* to facilitate breeding of *indica-japonica* hybrids.

Rice (*Oryza sativa* L.) is the preferred major staple crop for over half of the world population (FAO statistical databases, 2022). Historically, several major leaps in rice yield per unit land area have occurred due to improvement of breeding technologies and utilization of advanced cultivars. A major landmark in rice breeding is the utilization of the semi-dwarf gene *sd1* (encoding an enzyme essential for gibberellin biosynthesis) and breeding of lodging-resistant rice cultivars in the 1950s, leading to the "first-green revolution" in rice production (average rice yield increased from ~150 kg/mu to 300 kg/mu)[1]. The second landmark in rice breeding is development of hybrid rice in China in the 1970s, led by a group of scientists headed by professor Longping Yuan. Two-hybrid rice breeding technologies, the "three-line" (based on CMS-WA cytoplasmic male sterility) and the "two-line" (based on photoperiod- or

thermo-sensitive male sterility) hybrid systems, were developed sequentially in China in the 1970s and 1980s respectively, and are still widely used nowadays[2–4]. Three-line hybrid rice has a yield advantage of ~20–30% over the traditional inbred varieties, and two-line hybrid rice could confer ~5–10% additional yield increase over three-line hybrid rice[5]. To date, the large-scale application of commercial hybrid rice has increased total accumulated rice yield by over 0.6 billion tons, contributing significantly to securing global food supply[6,7].

*Indica* and *japonica* are two major genetically differentiated ecotypes of Asian cultivated rice. Currently, the majority of hybrid rice is intra-subspecific hybrid rice produced from crosses between *indica* varieties, with few inter-subspecific hybrid rice produced from crosses between *indica* and *japonica* subspecies (China Rice

[1]State Key Laboratory for Conservation and Utilization of Subtropical Agro-Bioresources, Guangdong Laboratory for Lingnan Modern Agriculture, South China Agricultural University, Guangzhou 510642, China. [2]Guangdong Key Laboratory of New Technology in Rice Breeding, Rice Research Institute, Guangdong Academy of Agricultural Sciences, Guangzhou 510640, China. [3]Life Science and Technology Center, China National Seed Group Co., LTD, Wuhan 430073, China. [4]These authors contributed equally: Yajun Gou, Yueqin Heng. ✉e-mail: whyang@scau.edu.cn; shenrongxin@scau.edu.cn

Data Center, https://www.ricedata.cn). Over the past few decades, yield of intra-subspecific *indica* hybrid rice has reached a plateau due to the limited genetic diversity of the parental lines[8]. As *indica-japonica* inter-subspecific hybrid rice has a yield advantage of ~15–30% over the intra-subspecific *indica/indica* hybrid rice, it is believed that developing more *indica-japonica* inter-subspecific hybrid rice is an essential route to boost rice production in the future. Nevertheless, production of *indica-japonica* hybrid rice is currently hindered by several technical bottlenecks, such as hybrid male sterility, prolonged life span, prone to lodging of the hybrids due to exaggerated plant height, and low yield of hybrid seed production. A major reason for low yield of hybrid seed production is asynchronous diurnal floret opening time (DFOT) of the *indica* and *japonica* parental lines. During production of *indica-japonica* inter-subspecific hybrid rice, *japonica* varieties are often used as the male sterile lines (maternal parent), whereas *indica* varieties are often used as the pollen donors (parental line). In general, *indica* rice has an earlier DFOT than *japonica* rice (average 1–3 h)[9]. This asynchronous DFOT of *indica* and *japonica* would severely reduce the efficiency of cross-pollination and large-scale hybrid seed production[10]. Therefore, genetic improvement and synchronization of DFOT of the *indica* and *japonica* parental lines are urgently needed for successful commercial production of *indica-japonica* inter-subspecific hybrid rice seeds.

Previous studies have revealed that lodicule, a small scale-like organ lying between the lemma/palea and stamens within the base of florets, plays an important role in controlling DFOT in rice. During floret opening, the lodicule cells absorb water and expand, pushing the lemma and palea apart to drive floret opening. After pollination, lodicule loses water and undergoes programmed cell death, leading to closure of the glumes[10–12]. The opening and closing processes of rice spikelets entail a dynamic change of turgor and osmotic pressure in the parenchymatous cells of the lodicule[13–15]. Mounting evidence indicated that the plant hormone jasmonic acid (JA) plays a key role in regulating floret opening and closure in rice and other monocotyledonous plant species such as wheat and rye[16–19]. In addition, several recent studies have reported that *Diurnal Flower Opening Time 1* (*DFOT1*) / *Early Morning Flowering1* (*EMF1*), which encodes a DUF642 domain-containing protein, negatively regulates rice DFOT through modulating methylesterification of lodicule cell wall pectin[11,20]. Despite the progress made in this field, the genetic basis and molecular mechanisms regulating DFOT in rice, especially the differentiation of DFOT in *indica* and *japonica*, have remained essentially unknown.

In this study, we identify an R2R3-MYB transcription factor OsMYB8 as a key regulator of asynchronous DFOT by performing a comparative time-course transcriptome analysis of lodicules in *indica* and *japonica*. We find that *OsMYB8* promotes floret opening in both *indica* and *japonica* rice through directly activating the expression of a JA biosynthesis gene *OsJAR1*, leading to elevated JA-Ile accumulation and thus altered expression of genes related to cell osmolality and cell wall remodeling in the lodicules. We also reveal that natural variation in the promoter sequences of *OsMYB8* confers higher expression level of *OsMYB8* in *indica*, thus higher accumulation of JA-Ile and earlier DFOT in *indica* as compared to *japonica*. Finally, we demonstrate that introgression of the *indica* *OsMYB8* allele into *japonica* rice could confer earlier DFOT in *japonica* rice. Our study not only provides a significant breakthrough into the understanding of the genetic network regulating asynchronous DFOT of *indica* and *japonica* rice but also provides effective genetic resources for genetic improvement of rice DFOT to facilitate hybrid breeding between *indica* and *japonica* subspecies.

## Results

### Diurnal floret opening time divergence in rice subspecies

To investigate the genetic variation for the DFOT trait in rice, we collected a rice diversity panel, including 12 *japonica* and 28 *indica* cultivars (Supplementary Data 1), and compared their DFOT in October 2019 in Guangzhou (Longitude, 113.27°; Latitude, 23.13°). The results showed that the *indica* cultivars exhibited the peak opening time at around 10:00–11:00 am, whereas the *japonica* cultivars had peak opening time at approximately 12:00–12:30 noon, indicating that the DFOT of *indica* cultivars was significantly earlier than that of *japonica* cultivars (Fig. 1a, Supplementary Data 1). Subsequently, the DFOT trait of a typical *indica* cultivar TianFengB (TFB) and a typical *japonica* cultivar ZhongHua11 (ZH11) was further analyzed in the early (June) and late (October) growing seasons of 2020 in Guangzhou. As expected, the DFOT of TFB was 1–2 h earlier than that of ZH11 in both the early and late seasons (Fig. 1b–e). Meanwhile, we carefully recorded the size and morphology of the lodicules of TFB and ZH11 at different time points under a stereomicroscope. The results showed that the lodicules of TFB swelled and reached the maximum size by 10:00 am, while the lodicules of ZH11 displayed a delay in swelling and did not reach the maximum size until 12:00 noon (Fig. 1f–h). Measurement of water content in the lodicules of TFB and ZH11 at different time points revealed a gradual increase of water content as the floret opening time approaching. Water content reached the highest at 10:00 am in the lodicules of TFB, but did not reach the highest until 12:00 noon in ZH11 (Fig. 1i). Taken together, these results indicate that the *indica* cultivars exhibit earlier DFOT than *japonica* cultivars, which could be attributed to advanced swelling of the lodicules in *indica* cultivars.

### Identification of *OsMYB8* as a key regulator of differential DFOT in *indica* and *japonica*

To explore the molecular basis underlying the differential lodicule swelling in *indica* and *japonica*, we performed a comprehensive comparative RNA-seq analysis of the lodicules of TFB and ZH11 at different time points, including 18:00 the day before floret opening (T18 in TFB, Z18 in ZH11), 3 h before opening in ZH11 (Z9, ~9:00 am), 1 h before opening (T9, ~9:00 am in TFB; Z11, ~11:00 am in ZH11) and undergoing peak floret opening time (TF, ~10:00 am in TFB; ZF, ~12:00 noon in ZH11) (Supplementary Fig. 1a, b). Correlation analysis of biological replicates from each time point showed very high correlation coefficients ($R^2 \geq 0.90$; Supplementary Fig. 1c). Thus, we calculated the average FPKM value of the replicates as the gene expression level for each sample. We further performed pairwise differential gene expression analyses between two consecutive time points and identified 6102 differentially expressed genes (DEGs) between T9 vs. T18, 6326 DEGs between TF vs. T9, 7164 DEGs between Z9 vs. Z18, 2972 DEGs between Z11 vs. Z9, 7012 DEGs between ZF vs. Z11 and 7430 DEGs between T9 vs. Z9 (*P*-value < 0.05, absolute log$_2$FC (fold change) ≥ 1; Supplementary Fig. 1d; Supplementary Data 2). Gene Ontology (GO) term enrichment analysis revealed that biological processes and molecular pathways related to cell osmolality and cell wall remodeling, such as sugar transport, water channel activity, JA synthesis and signaling, calcium signaling and ethylene signaling were enriched in one or more datasets (Supplementary Fig. 2). Principal component analysis (PCA) showed that the dataset of T18 was grouped together with that of Z18. Likewise, the dataset of TF was grouped together with that of ZF (Fig. 2a). In congruent with earlier DFOT in TFB, the T9 time point was grouped together with Z11 but not Z9, suggesting that transcriptional activities related to floret opening process are advanced in TFB, compared to ZH11 (Fig. 2a).

Considering that the time point of 1 h before opening is the crucial stage for floret opening[10,21], we speculated that genes responsible for differentiated DFOT in *indica* and *japonica* might show characteristic expressional changes in T9 and Z11, and likely be up-regulated in the

datasets of T9 vs. Z9. Therefore, we first performed k-means clustering analysis of all expressed genes and identified a total of 20 clusters with distinct gene expression patterns (Supplementary Fig. 1e). Among them, the Cluster18 genes showing peak expression at T9 and Z11 were selected for further detailed analyses. Next, we performed an overlap analysis of Cluster18 genes with up-regulated DEGs of T9 vs. Z9. A total of 251 overlapping genes were identified (Fig. 2b). GO term analysis revealed an enrichment of DNA binding and transcription factor activity (Fig. 2c). We further noted that among the 11 overlapping genes encoding transcription factors, *LOC_Os01g45090* (*OsMYB8*), encoding an R2R3-MYB transcription factor, exhibited the highest expression level and was gradually up-regulated in the lodicules before floret opening (Fig. 2d). Neighbor-joining tree analysis showed that *OsMYB8* is homologous to *AtMYB21/24*, which are involved in JA signaling and regulating stamen development in *Arabidopsis*[22] (Supplementary Fig. 3). The RiceXPro data and reverse transcriptional quantitative PCR (RT-qPCR) analyses showed that *OsMYB8* was preferentially expressed in lemma, palea, stamen, stigma, and lodicule, but with low expression level in root, stem, leaf, and young panicle (Supplementary Fig. 4a, b). Histochemical staining analysis of transgenic lines expressing the *pOsMYB8::GUS* (β-glucuronidase) reporter gene also revealed a similar expression pattern (Supplementary Fig. 4c). RT-qPCR analysis showed that the expression level of *OsMYB8* was lower in the lodicules of TFB and ZH11 at the day before floret

opening, and then gradually increased towards opening time, and decreased thereafter. Additionally, we found that the expression level of *OsMYB8* in *indica* lodicules was significantly higher than that in *japonica* lodicules before floret opening (Supplementary Fig. 5). Moreover, RT-qPCR analysis verified that the expression levels of *OsMYB8* in the lodicules of TFB at 9:00 am (the time before floret opening) were about 2-fold of that in ZH11 at the same time points (Fig. 2e). These observations together suggest that the elevated expression level and an earlier peak expression time of *OsMYB8* in TFB compared with that in ZH11 may contribute to the earlier floret opening time in *indica* varieties. Thus, we selected *OsMYB8* as a candidate gene regulating the differentiated DFOT in *indica* and *japonica* subspecies.

## *OsMYB8* promotes floret opening in rice

To explore the role of *OsMYB8* in DFOT regulation, we knocked out the *OsMYB8* gene in both ZH11 and TFB backgrounds using the CRISPR/Cas9 technology. Two verified independent *OsMYB8* knockout lines in ZH11 (*Osmyb8^ZH#1* and *Osmyb8^ZH#2*) and TFB background (*Osmyb8^TF#1* and *Osmyb8^TF#2*) were selected for detailed phenotypic analysis (Fig. 3a, Supplementary Fig. 6). The florets of ZH11 started opening at around 10:30 am, and achieved peak opening time by 11:30 am, whereas the *Osmyb8^ZH* mutants did not achieve peak opening time until 12:00 noon, about 0.5 h later than ZH11 (Fig. 3b, c). Similarly, the

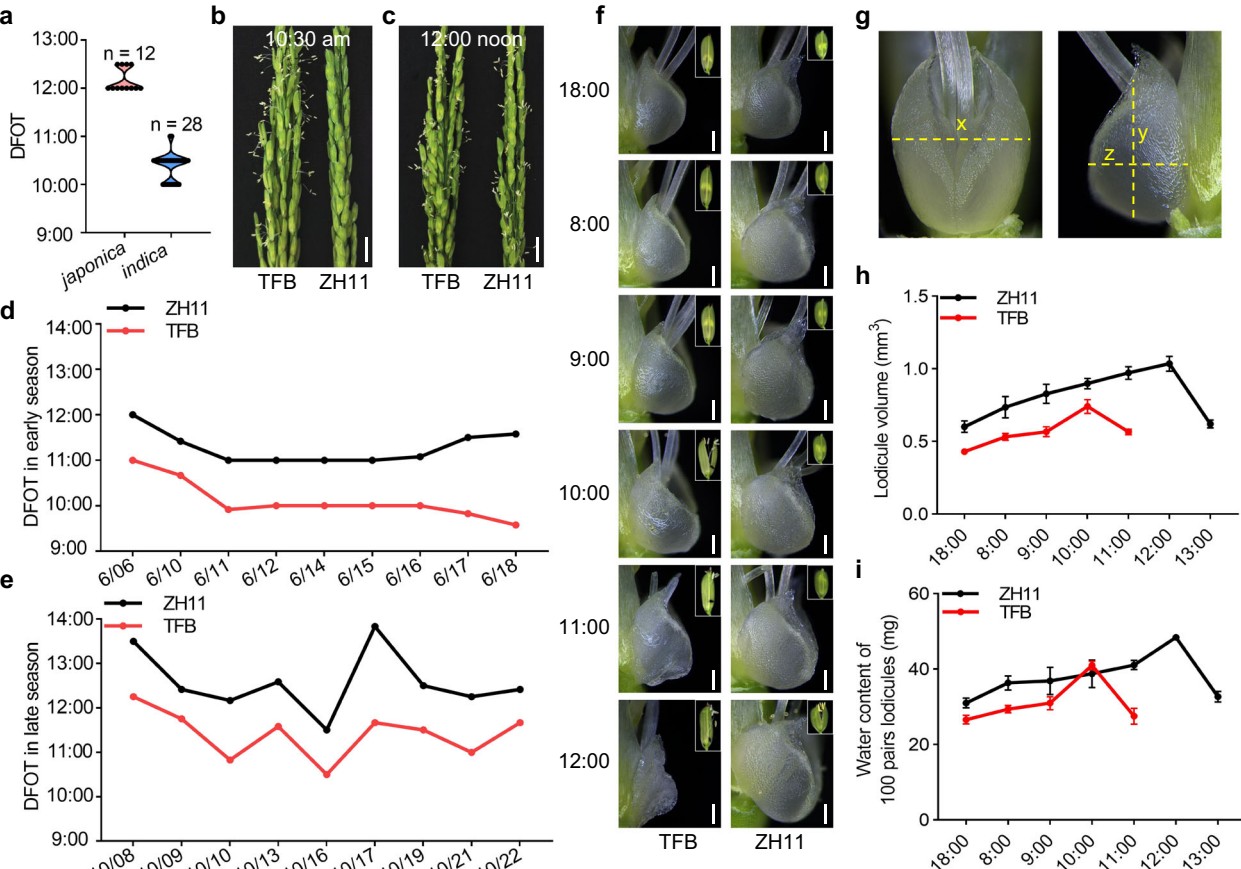

**Fig. 1 | DFOT divergence in rice subspecies. a** Diurnal floret opening time (DFOT) of *japonica* cultivars (*n* = 12 accessions) and *indica* cultivars (*n* = 28 accessions) in October 2019 in Guangzhou. **b, c** Comparison of panicles undergoing floret opening in the *indica* cultivar TFB and *japonica* cultivar ZH11 at 10:30 am (**b**) and 12:00 noon in October 2020 in Guangzhou (**c**). Scale bars, 1 cm. **d, e** The DFOT of TFB and ZH11 in June 2020 in Guangzhou (**d**) and October 2020 in Guangzhou (**e**). **f** Lodicule morphology of TFB and ZH11 at different time points in October 2020 in Guangzhou. Scale bars, 250 μm. The boxed areas indicate the corresponding

florets. (*n* = 10 lodicules). **g** Viewing the lodicule as an ellipsoid, its x, y, and z-axes are shown in the diagram. **h** Quantitative comparison of lodicules volume in ZH11 and TFB at different time points. Lodicule volume was calculated using the ellipsoid volume formula (V = 4πabc/3). **a**, **b**, and **c** correspond to the semi-axes along the x, y, and z-axes shown in g. Values are means ± SEM. (*n* = 10 lodicules). **i** Analysis of the water content of 100 pairs lodicules of ZH11 and TFB at different time points. Values are means ± SEM. (*n* = 3 biological replicates). Source data are provided as a Source Data file.

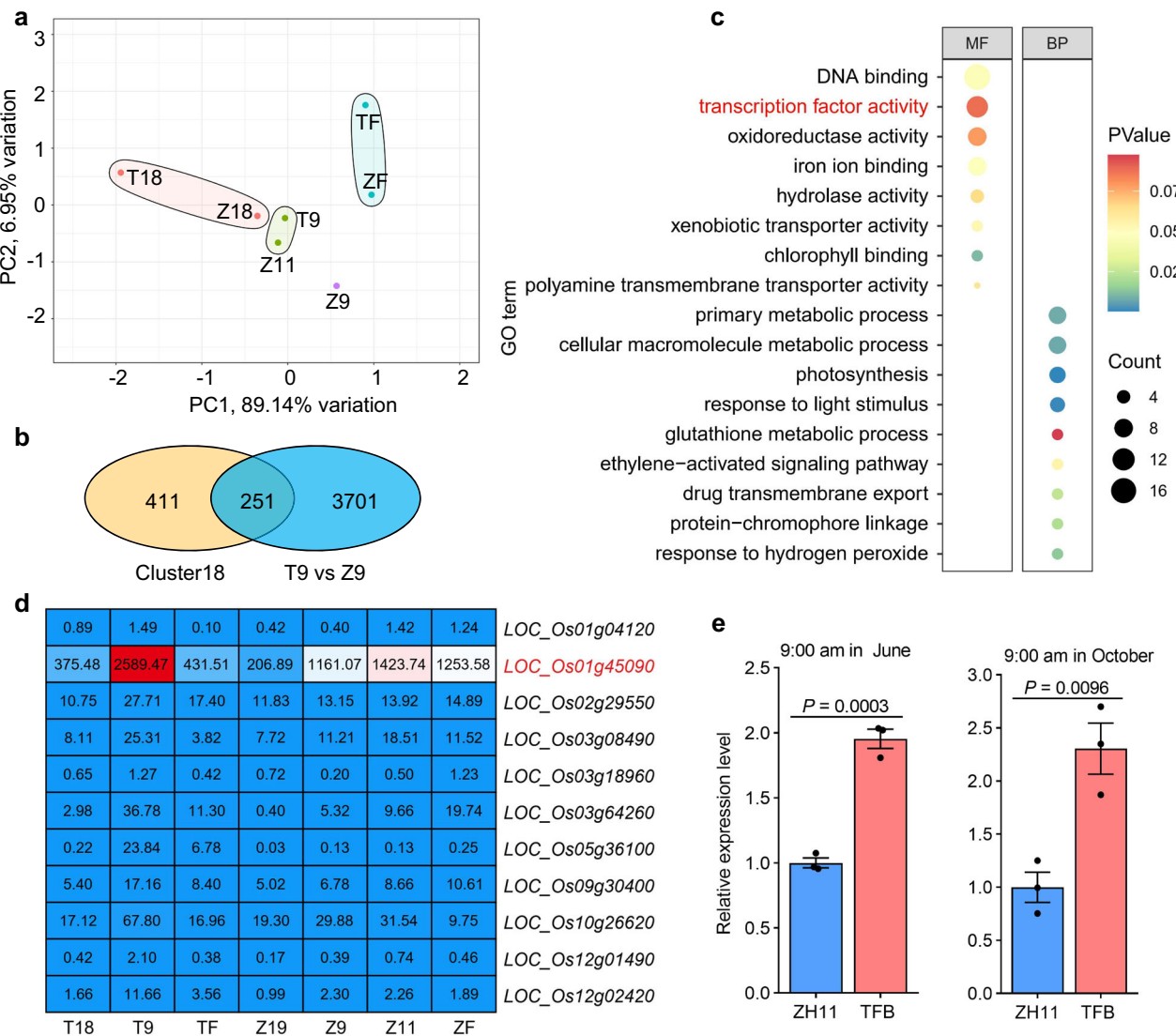

**Fig. 2 | Identification of *OsMYB8* by comparative transcriptomic analysis of lodicules between *indica* and *japonica*. a** Principal component analysis (PCA) of the transcriptome datasets of lodicules between TFB and ZH11 at different time points. T18 and Z18 indicate 18:00 the day before floret opening; T9 and Z9 indicate 9:00 am (1 h and 3 h before floret opening in TFB and ZH11, respectively); Z11 indicate 11:00 am (1 h before opening in ZH11); TF and ZF indicate the time undergoing peak floret opening time (-10:00 am in TFB; -12:00 noon in ZH11). **b** Venn Diagram showing the number of overlapping genes between the Cluster18 genes and up-regulated DEGs from T9 vs Z9. **c** Gene Ontology (GO) enrichment

analysis of 251 overlap genes in (**b**). MF indicates molecular function, BP indicates biological process. The term of "transcription factor activity" is highlighted in red. **d** Expression analysis of 11 transcription factors selected through GO analysis in (**c**). The numbers in the heatmap represent the average FPKM values. **e** Reverse transcriptional quantitative PCR (RT-qPCR) analysis of *OsMYB8* expression level in the lodicules of TFB and ZH11 at 9:00 am in June and October in 2021. Values are mean ± SEM. (*n* = 3 biological replicates). Significance is evaluated by the two-sided Student's *t*-test at each time point, and *P* values are indicated. Source data are provided as a Source Data file.

peak opening time of the *Osmyb8*<sup>TF</sup> mutants was about 1 h later than TFB (Fig. 3e, f). In addition, given that *OsMYB8* showed higher expression level in TFB than in ZH11, we also generated two transgenic lines expressing *OsMYB8* of TFB (including the coding sequence and 2-kb upstream promoter sequence) in the ZH11 background (*OsMYB8*<sup>TF</sup>/ZH11#1 and *OsMYB8*<sup>TF</sup>/ZH11#2) (Fig. 3h, i). As shown in Fig. 3j, the peak opening time of the *OsMYB8*<sup>TF</sup>/ZH11 plants occurred -0.5–1 h earlier than ZH11 (Fig. 3j, k). The daily number of opening florets of both the *Osmyb8* mutants and *OsMYB8*<sup>TF</sup>/ZH11 lines was comparable to that of the wild-type plants (Fig. 3d, g, l). Additionally, both the *OsMYB8* knockout lines in ZH11 and TFB backgrounds exhibited normal spikelet and stamen development, and their anthers also dehisced normally (Supplementary Fig. 7a–c). Besides, $I_2$-KI (potassium iodide) solution staining experiments showed that the pollen fertility of the *Osmyb8* mutants were comparable to that of the WT plants (Supplementary Fig. 7d, e). Moreover, seed setting rates of

*Osmyb8*<sup>ZH</sup>#1 and *Osmyb8*<sup>ZH</sup>#2 were comparable to ZH11, while *Osmyb8*<sup>TF</sup>#1 and *Osmyb8*<sup>TF</sup>#2 showed slightly lower seed setting rates compared to TFB (Supplementary Fig. 7f). These results demonstrated that *OsMYB8* primarily acts to promote rice floret opening but confer minimal influence on the pollen development and spikelet fertility.

**Identification of the genome-wide direct targets of OsMYB8**

Consistent with *OsMYB8* encoding an R2R3-MYB transcription factor, the OsMYB8-GFP fusion protein was localized to the nucleus in rice protoplasts (Fig. 4a). Moreover, transcriptional activity analysis of OsMYB8 in yeast revealed that OsMYB8 is a transcriptional activator and the activation domain is located at its C-terminus (Fig. 4b).

To identify the direct downstream targets of OsMYB8, we first performed DNA affinity purification sequencing (DAP-seq) to unravel its genome-wide binding sites. Purified recombinant GST-OsMYB8 fusion protein (expressed in *E. coli*) was used to affinity purify the

sheared genomic DNA of 14-day old ZH11 seedlings, followed by deep sequencing (Supplementary Fig. 8a). In total, 27,764 binding peaks located within the regulatory regions of 20,832 genes were detected

(Supplementary Data 3), 70.74% of which were distributed in the promoter or intergenic regions, 13.26% and 11.82% were in the exon and the intron regions respectively, while 4.18% was in the 5'UTR and 3'

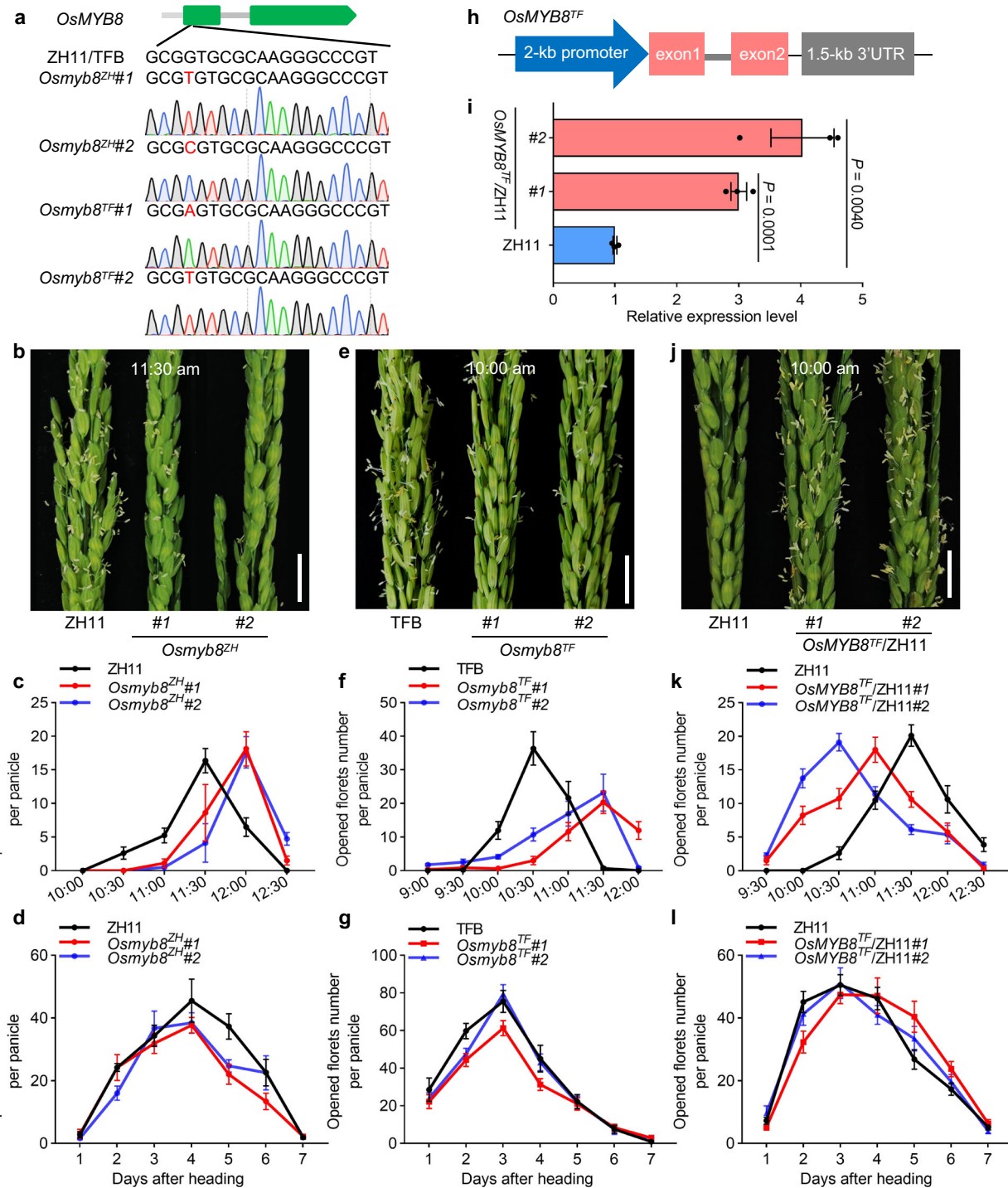

**Fig. 3 | *OsMYB8* positively regulates rice DFOT. a** Creation of the *Osmyb8* mutants using the CRISPR/Cas9 genome editing approach. *Osmyb8* mutants in ZH11 and TFB backgrounds were named as *Osmyb8^ZH* and *Osmyb8^TF* respectively. The upper panel shows a schematic diagram of the *OsMYB8* gene bearing the CRISPR/Cas9 target site. The mutation site was indicated in red. **b**, **e**, **j** Comparison of panicles in ZH11 and *Osmyb8^ZH* mutants at 11:30 am (**b**) TFB and *Osmyb8^TF* mutants at 10:00 am (**e**) ZH11 and *OsMYB8^TF*/ZH11 lines at 10:00 am in June 2021 in Guangzhou (**j**). Scale bars, 1 cm. **c**, **f**, **k** Number of opened florets per panicle in ZH11 and *Osmyb8^ZH* mutants (**c**), TFB and *Osmyb8^TF* mutants (**f**), and ZH11 and *OsMYB8^TF*/ZH11 lines (**k**) at different time points of the day in June 2021 in Guangzhou. Values are means ± SEM. (*n* = 8

panicles). **d**, **g**, **l** Number of opened florets in ZH11 and *Osmyb8^ZH* mutants (**d**), TFB and *Osmyb8^TF* mutants (**g**), and ZH11 and *OsMYB8^TF*/ZH11 lines (**l**) at different days after heading in June 2021 in Guangzhou. Values are means ± SEM. (*n* = 10 panicles). **h** Schematic diagram of the vector structure used for constructing *OsMYB8^TF*/ZH11 transgenic plants. **i** RT-qPCR analysis of *OsMYB8* transcripts levels in the lodicules of the *OsMYB8^TF*/ZH11 transgenic lines. ZH11 was used as a negative control. Values are means ± SEM. (*n* = 3 biological replicates). Significance is evaluated by the two-sided Student's *t*-test, and *P* values are indicated. Source data are provided as a Source Data file.

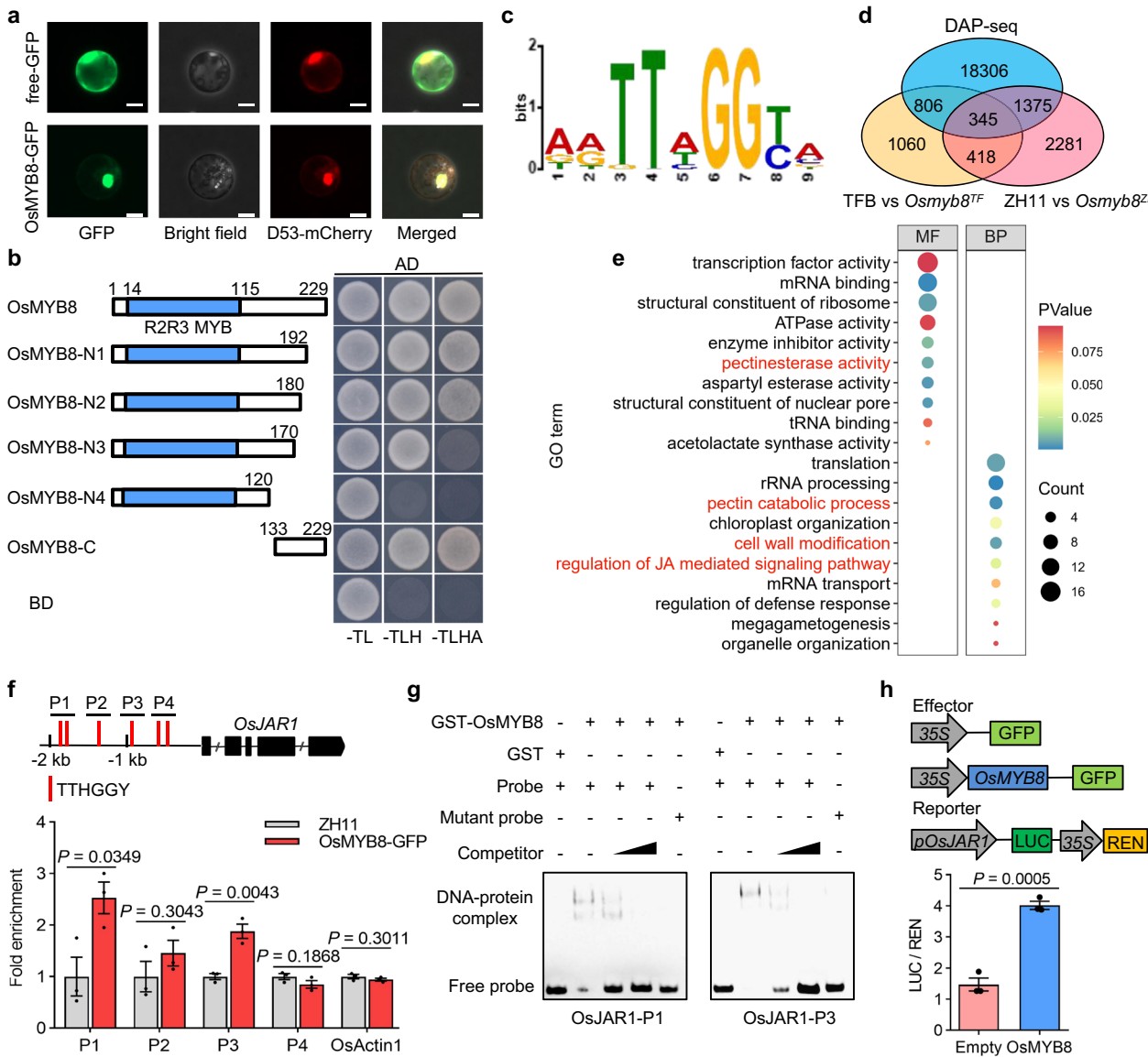

**Fig. 4 | Identification of the genome-wide direct targets of OsMYB8. a** Nuclear localization of OsMYB8-GFP in rice protoplasts. D53-mCherry was used as a nuclear marker. Scale bars, 20 μm. **b** Yeast two-hybrid assay showing the transcriptional activity of OsMYB8. Left is the diagrams of full-length and truncated OsMYB8 proteins, which were fused with the DNA-binding domain (BD). **c** The core binding motif of OsMYB8 protein identified using MEME-ChIP. **d** Venn diagrams showing a comparison of DAP-seq binding genes with the identified DEGs in *Osmyb8* mutants by RNA-seq. **e** GO enrichment analysis of 345 overlap genes in (**d**). MF indicates molecular function, BP indicates biological process. **f** ChIP-qPCR analysis showing that OsMYB8 binds to the *OsJAR1* promoter in vivo. The upper panel is a diagram of *OsJAR1* promoter with the indicated regions used for detection by ChIP-qPCR. The red lines indicate the position of "TTHGGY" motifs in the *OsJAR1* promoter regions. Immunoprecipitation was performed with anti-GFP antibody, ZH11 was used as a

negative control. (*n* = 3 technical replicates). **g** EMSA assay showing that GST-OsMYB8 recombinant protein directly binds to the "TTHGGY" motif-containing regions of the *OsJAR1* promoter. Unlabeled probes were used as competitors. GST was used as a negative control. **h** Transient dual-LUC assay showing that OsMYB8 induces the transcriptions of *OsJAR1* promoter in rice protoplasts. The upper panel is diagrams of various constructs used in the transient expression assay. The expression level of REN was used as an internal control. The LUC/REN ratio represents the relative activity of the *OsJAR1* promoter. (*n* = 3 technical replicates). The Values in (**f**) and (**h**) are means ± SEM. Significance is evaluated by the two-sided Student's *t*-test, and *P* values are indicated. The subcellular localization experiments in (**a**) and the EMSA experiments in (**g**) were independently repeated three times with similar results. Source data are provided as a Source Data file.

UTR regions (Supplementary Fig. 8b). By using the MEME Suite, a representative DNA binding motif of MYB proteins with a core sequence of "TTHGGY" (H indicates A/T/G, Y indicates T/C) was significantly enriched among the OsMYB8 binding regions[23] (Fig. 4c). To verify the reliability of our DAP-seq result, we randomly selected 3 putative target genes, and further carried out electrophoretic mobility shift assays (EMSAs) using native promoter probes harboring the wild-type "TTHGGY" motif or its mutated forms "TTHAAY". The results showed that OsMYB8 could directly bind to the native promoter probes, whereas mutations in the "TTHGGY" motif largely abolished

the binding of OsMYB8 (Supplementary Fig. 8c), thus confirming that OsMYB8 could directly bind to the "TTHGGY" motif.

To narrow down the potential direct targets of OsMYB8 for regulating rice DFOT, we performed RNA-seq of the lodicules of TFB and *Osmyb8^TF* collected at 9:00 am or lodicules of ZH11 and *Osmyb8^ZH* collected at 10:00 am. Correlation analysis showed very high correlation coefficients within three biological replicates for each group (Supplementary Fig. 8d). Among them, 2629 DEGs were identified between TFB vs. *Osmyb8^TF*, and 4419 DEGs were identified between ZH11 vs. *Osmyb8^ZH*, respectively (*P*-value < 0.05, absolute $\log_2$FC ≥ 1;

Supplementary Fig. 8e, Supplementary Data 4, 5). By overlapping analysis of the DEGs and the OsMYB8 binding genes, we identified 345 potential genes that were differentially expressed in *Osmyb8* lodicules and directly bound by OsMYB8 (Fig. 4d). GO term analysis revealed significant enrichment of several molecular functions known to be related to lodicule swelling and closure, such as cell wall modification and regulation of JA-mediated signaling pathway (Fig. 4e, Supplementary Data 6). Among them, the *OsJAR1* gene, which encodes an enzyme essential for the conversion of JA to active JA (JA-Ile)[24], was down-regulated in the lodicules of the *Osmyb8* mutants, but up-regulated in the lodicules of the *OsMYB8^TF*/ZH11 lines (Supplementary Fig. 9a–c). RT-qPCR analysis revealed that similar to *OsMYB8*, the expression level of *OsJAR1* in the lodicules of TFB and ZH11 increases as the floret opening time approaches (Supplementary Fig. 9d).

To verify that *OsJAR1* is a direct target of OsMYB8, we analyzed the promoter region of *OsJAR1*, and identified a set of OsMYB8 binding motifs (TTHGGY) in the 2-kb region upstream of the transcription start site (Fig. 4f). To examine the binding capacity of OsMYB8 to the *OsJAR1* promoter in vivo, we generated two transgenic lines with stable expression of *OsMYB8-GFP* driven by the native *OsMYB8* promoter of TFB (Supplementary Fig. 10a–c). We also found that the *OsMYB8-GFP* lines exhibited about 1 h earlier DFOT than ZH11 (Supplementary Fig. 10d, e), suggesting that the OsMYB8-GFP fusion protein is biologically functional. ChIP-qPCR assays using florets of the *OsMYB8-GFP#2* transgenic line showed that anti-GFP antibody could specifically precipitate the P1 and P3 fragments containing the OsMYB8 binding motif in the *OsJAR1* promoter (Fig. 4f). EMSAs also demonstrated that GST-OsMYB8 could directly bind to the P1 and P3 fragments of the *OsJAR1* promoter (Fig. 4g). In addition, transient expression assay in rice protoplasts showed that *LUC* expression driven by the native *OsJAR1* promoter was increased significantly when the reporter was co-transformed with the OsMYB8 effector in rice protoplasts (Fig. 4h). Taken together, our findings suggest that OsMYB8 directly activates *OsJAR1* transcription.

### OsMYB8 genetically acts upstream of OsJAR1 to regulate rice DFOT

To test whether *OsJAR1* regulates rice DFOT, we performed targeted mutagenesis of *OsJAR1* in the ZH11 background using the CRISPR/Cas9 technology. Two independent knockout lines (*Osjar1#1* and *Osjar1#2*) were obtained for phenotypic analysis (Fig. 5a). We counted the number of opened florets at different time points of ZH11 and *Osjar1* mutants in June 2022 in Guangzhou, and found that in contrast to ZH11, which had a peak DFOT at around 11:30 am, the *Osjar1* mutants displayed a scattered floret opening phenotype (random floret opening throughout the day), indicating that *OsJAR1* is required for proper floret opening (Fig. 5b, c). As expected, measurement of JA-Ile in the lodicules of ZH11, *Osjar1* mutant and *Osmyb8^ZH* showed that the JA-Ile content in the lodicules of *Osjar1* and *Osmyb8^ZH* mutant was significantly lower than that of ZH11 (Fig. 5d). In contrast, the JA-Ile content of *OsMYB8^TF*/ZH11 was significantly higher than that of ZH11 (Fig. 5e). These data together suggest that *OsMYB8* regulates JA-Ile content in lodicules to promote floret opening in rice.

To further determine the genetic relationship between *OsJAR1* and *OsMYB8*, we introduced the *OsJAR1* coding sequence driven by the *OsMYB8* promoter of TFB (which possesses high transcriptional activity in lodicules, Fig. 2d) into the *Osmyb8^ZH* mutant background (Fig. 5f). Two independent transgenic lines (*OsJAR1^com#1* and *OsJAR1^com#2*) with higher *OsJAR1* expression level in lodicules than that of ZH11 and *Osmyb8^ZH* exhibited an intermediate peak DFOT between ZH11 and *Osmyb8^ZH* (Fig. 5g–i), suggesting that up-regulation of *OsJAR1* expression could partially complement the delayed DFOT of *Osmyb8^ZH*. Moreover, we generated *OsMYB8^TF*/*Osjar1* line by crossing *OsMYB8^TF*/ZH11 with an *Osjar1* line and found that the *OsMYB8^TF*/*Osjar1* line displayed a scattered floret opening phenotype, similar to *Osjar1*

(Fig. 5j, k). These observations support the placement of *OsJAR1* downstream of *OsMYB8* to regulate DFOT in rice.

As genes related to cell osmolality and cell wall remodeling play a predominant role in DFOT regulation[10,21] (Supplementary Fig. 2), we next tested whether dysfunction of *OsJAR1* could influence their expressions. As expected, we found that a series of differentially expressed genes associated with carbohydrate metabolic process, sugar transport, cell wall organization, and water channel activity showed altered expression in the transcriptome data of TFB vs. *Osmyb8^TF* and ZH11 vs. *Osmyb8^ZH* (Supplementary Fig. 11a), and most of the DEGs were down-regulated in lodicules of both *Osmyb8^TF* and *Osmyb8^ZH*. Remarkably, four *Pectin Methylesterase* genes (*OsPME12/22/23/29*) were up-regulated, in line with the potential negative role of this gene family in DFOT regulation[11]. We further performed RT-qPCR analyses to examine the expression changes of these genes in the lodicules of the *Osjar1* mutants. As expected, a large portion of genes exhibited similar changes to that of the *Osmyb8* mutants. For example, expression of *OsAmy2*, *Os4BGlu*, *OsSWEET11/15*, *OsEXPB7*, *OsXTH16*, *OsPG17* and *OsNIP1;1* was significantly decreased, while expression of *OsPME23/29* was significantly increased (Supplementary Fig. 11b, c). We further analyzed soluble sugar content in the lodicules of ZH11 and *Osmyb8^ZH* collected at 10:00 am, and the result showed that the levels of sucrose, fructose and total soluble sugars were decreased in the lodicules of *Osmyb8^ZH* (Supplementary Fig. 12). These results suggest that *OsMYB8-OsJAR1* module likely influence floret opening by modulating the expression of genes related to lodicule hydration and expansion.

### Natural variation in *OsMYB8* promoter confers DFOT divergence in *japonica* and *indica*

To look for the natural variation underlying differential expression of *OsMYB8* in *indica* and *japonica* rice, we conducted sequence analysis of the coding region and 2-kb promoter sequences of *OsMYB8* between TFB and ZH11. The results showed that there are six SNPs (−1176, −1244, −1286, −1469, −1550 and −1871) in the promoter and one synonymous SNP in the coding region (Fig. 6a). Based on the six SNPs, we performed haplotype analysis of the *OsMYB8* 2-kb promoter using 1973 *indica* (*Ind*), 767 *temperate japonica* (*TeJ*), 504 *tropical japonica* (*TrJ*), and 269 *Aus*[25] (Supplementary Data 7). A total of three haplotypes (Hap1–3) was identified in these rice accessions. Hap1 and Hap2 differed in six SNPs, while Hap3 differed from Hap1 at four SNPs (−1286, −1469, −1550, and −1871), and Hap2 at two SNPs (−1176 and −1244) (Fig. 6a). Hap1 was mainly present in the *Ind* accessions (including TFB). In contrast, 671 out of 767 (-88.0%) *TeJ* accessions (including ZH11) carried Hap2. Hap3 was mostly detected in *Ind*, *TrJ*, and *Aus* (Fig. 6b). To investigate whether *OsMYB8* has undergone selection during rice domestication, we surveyed the nucleotide diversity (π) across the *OsMYB8* genomic region using 295 *Ind* and 91 *TeJ* from a rice core collection[26] and 185 *O. rufipogon* (*Ruf*)[27] (Supplementary Data 8). The nucleotide diversity of the *OsMYB8* 2-kb promoter region was extremely low in *Ind* (π = 0.00074) and *TeJ* (π = 0.00076), as compared with that of *Ruf* (π = 0.0053), and was also much lower than that of the whole genome of *O. sativa* (π = 0.0024)[28] (Fig. 6c, Supplementary Data 9). Meanwhile, the fixation index ($F_{ST}$) value around *OsMYB8* exhibited a higher level of differentiation between the *Ind* and *TeJ* populations (*Ind_TeJ*) (Fig. 6d, Supplementary Data 10). These results indicate that *OsMYB8* has undergone divergent selection in the *indica* and *japonica* subspecies.

To further evaluate the functional difference between *OsMYB8^Hap1* and *OsMYB8^Hap2*, we analyzed dozens of accessions with Hap1 and Hap2, respectively (Supplementary Data 8). We found that the Hap1-carrying accessions exhibited earlier DOFT compared to the Hap2-carrying accessions (Fig. 6e). Furthermore, the Hap1-carrying accessions showed higher levels of *OsMYB8* gene expression and JA-Ile level

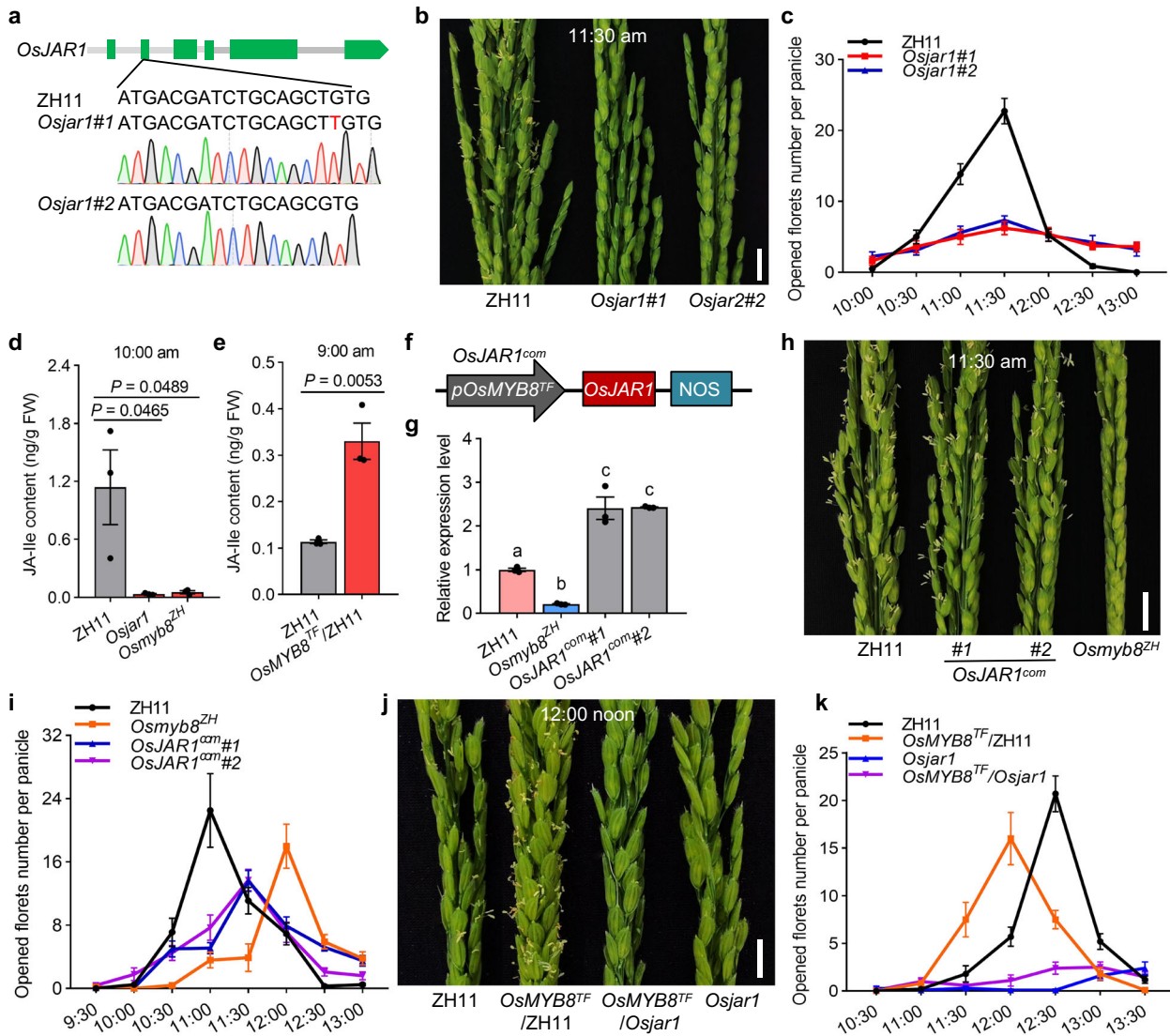

**Fig. 5 | *OsJAR1* influences JA-Ile content in lodicule to regulate rice DFOT.**
**a** Creation of the *Osjar1* mutants in ZH11 background using the CRISPR/Cas9 gen-ome editing approach. The mutation site was indicated in red. **b** Comparison of panicles in ZH11 and *Osjar1* mutants at 11:30 am in June 2022 in Guangzhou. Scale bars, 1 cm. **c** Number of opened florets per panicle in ZH11 and *Osjar1* mutants at different time points of the day. Values are mean ± SEM. (*n* = 10 panicles). **d**, **e** JA-Ile content in lodicules at 10:00 am of ZH11, *Osjar1,* and *Osmyb8^{ZH}* (**d**) at 9:00 am of ZH11 and *OsMYB8^{TF}*/ZH11(**e**). Values are means ± SEM. (*n* = 3 biological replicates). Significance is evaluated by the two-sided Student's *t*-test, and *P* values are indi-cated. **f** Schematic diagram of the vector structure used for constructing *OsJAR1^{com}* materials. The *pOsMYB8^{TF}* means the promoter was amplified from TFB. **g** Relative

expression level of *OsJAR1* in the lodicules of ZH11, *OsJAR1^{com}* and *Osmyb8^{ZH}*. Values are means ± SEM. (*n* = 3 biological replicates). Letters above the bars indicate sig-nificant differences (*P* < 0.05), as evaluated by one-way ANOVA with Tukey's mul-tiple comparisons test. **h**, **j** Comparison of panicles in ZH11, *OsJAR1^{com}* and *Osmyb8^{ZH}* at 11:30 am in June 2022 in Guangzhou (**h**) and in ZH11, *OsMYB8^{TF}*/ZH11, *OsMYB8^{TF}*/ *Osjar1* and *Osjar1* at 12:00 noon in October 2022 in Guangzhou (**j**). Scale bars, 1 cm. **i** Number of opened florets in ZH11, *OsJAR1^{com},* and *Osmyb8^{ZH}* at different time points of the day in June 2022 in Guangzhou. Values are mean ± SEM. (*n* = 10 panicles). **k** Number of opened florets in ZH11, *OsMYB8^{TF}*/ZH11, *OsMYB8^{TF}*/*Osjar1*, and *Osjar1* at different time points of the day in October 2022 in Guangzhou. Values are mean ± SEM. (*n* = 10 panicles). Source data are provided as a Source Data file.

(Fig. 6f, g). Moreover, transient transcriptional activation experiments showed that the *OsMYB8^{Hap1}* promoter possessed higher transcrip-tional activity than the *OsMYB8^{Hap2}* promoter in rice protoplast (Fig. 6h). These findings suggest that the sequence variations between Hap1 and Hap2 might cause differential expression of the *OsMYB8*, thus conferring earlier DFOT in the Hap1-carrying varieties as com-pared to the Hap2-carrying varieties. To consolidate this notion, we introduced the TFB *OsMYB8* allele (*OsMYB8^{TF}*) containing the 2-kb promoter and intact coding region into the *Osmyb8^{ZH}* mutant to gen-erate complementary plants. Two independent transgenic lines with single copy insertion of *OsMYB8^{TF}* (*OsMYB8^{TF}*/*Osmyb8^{ZH}*#1, *OsMYB8^{TF}*/ *Osmyb8^{ZH}*#2) were selected to be further analyzed (Supplementary Fig. 13). RT-qPCR analysis showed that the expression levels of *OsMYB8*

in lodicules of *OsMYB8^{TF}*/*Osmyb8^{ZH}* plants were about 2-fold that of ZH11 and *Osmyb8^{ZH}* mutant (Fig. 6i), similar to the expression differ-ence of *OsMYB8* between TFB and ZH11 (Fig. 2d, e). Consistently, the expression level of *OsJAR1* was higher in *OsMYB8^{TF}*/*Osmyb8^{ZH}* than in ZH11 and the *Osmyb8^{ZH}* mutant, and the JA-Ile content in the lodicules of *OsMYB8^{TF}*/*Osmyb8^{ZH}* was significantly higher than that of ZH11 (Supplementary Fig. 14). Phenotypic analysis showed that the *OsMYB8^{TF}*/*Osmyb8^{ZH}* plants exhibited about 0.5 and 1 h earlier DFOT than ZH11 and the *Osmyb8^{ZH}* mutant, respectively (Fig. 6j, k). These results together support the notion that the natural variation in the *OsMYB8* promoter contributes to the differential expression of *OsMYB8*, and thus divergence of DFOT in *indica* and *japonica* subspecies.

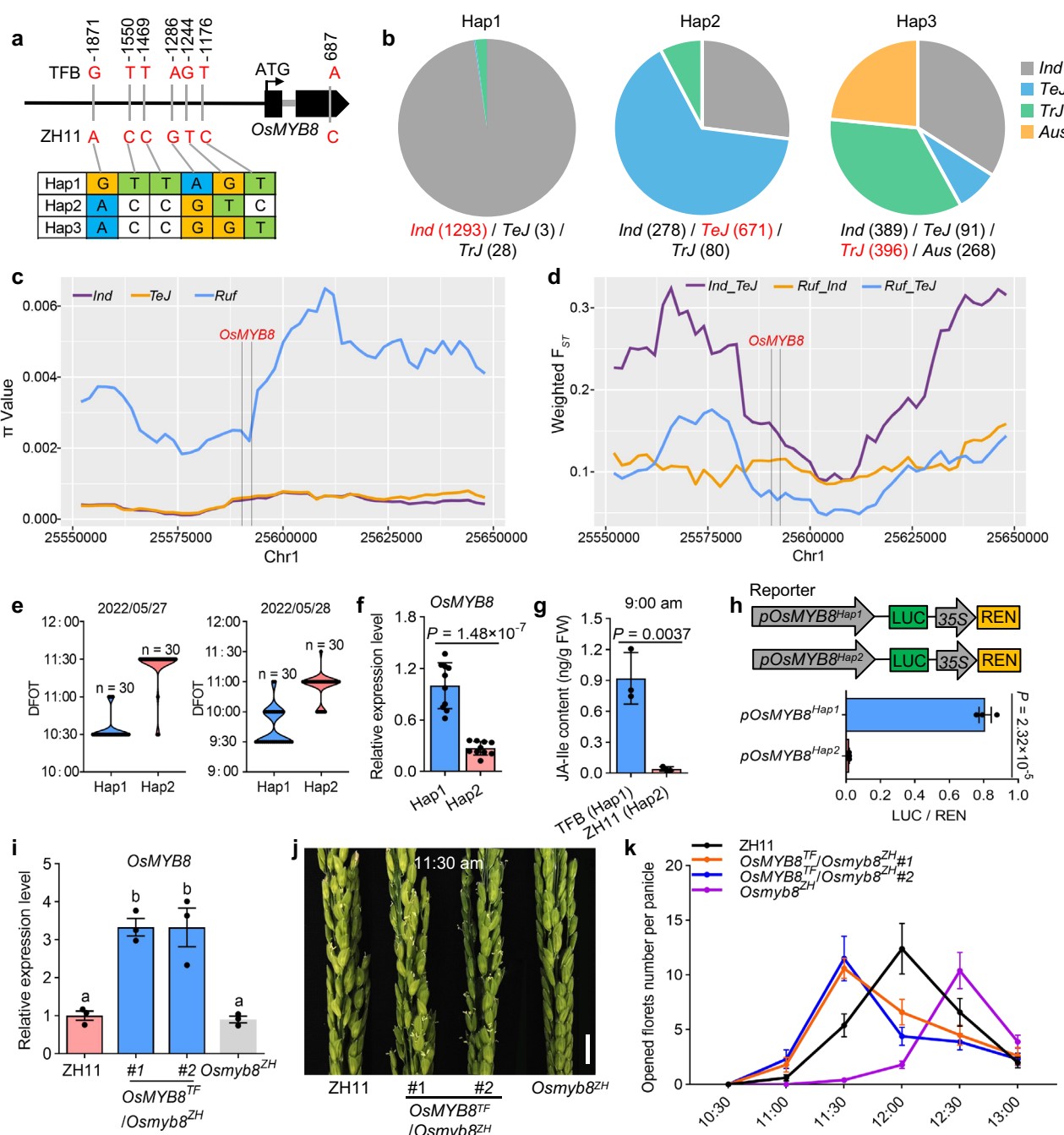

**Fig. 6 | Natural variation in *OsMYB8* promoter confers DFOT divergence in *japonica* and *indica*. a** Haplotype analysis of *OsMYB8* promoter in the 3513 rice germplasms. Nucleotide variations in the 2-kb promoter of *OsMYB8* were shown. **b** Distribution frequency of the three *OsMYB8* haplotypes in diverse Asian cultivated rice accession. The haplotype with the largest number was highlighted in red. **c** Nucleotide diversity (π) of a 100-kb genomic region surrounding *OsMYB8* in the *indica* (*Ind*), *temperate japonica* (*TeJ*) and *O. rufipogon* (*Ruf*). The regions (Chr1: 25,590,725–25,592,725) between two black vertical lines indicates the position of *OsMYB8* promoter. **d** $F_{ST}$ values of *TeJ_Ind*, *Ruf_TeJ*, and *Ruf_Ind* in a 100-kb genomic region surrounding *OsMYB8*. The region (Chr1: 25,590,725–25,592,725) between two black vertical lines indicates the position of *OsMYB8* promoter. **e** The DFOT of the rice accessions with Hap1 and Hap2 in May 2022 in Guangzhou. (*n* = 30 accessions). **f** Relative expression levels of *OsMYB8* in lodicules of the rice accessions with Hap1 and Hap2, respectively. (*n* = 10 accessions). **g** JA-Ile content in TFB and ZH11 lodicules at 9:00 am. (*n* = 3 biological replicates). **h** Transient dual-luciferase assays showing the transcriptional activity of *pOsMYB8*$^{Hap1}$ and *pOsMYB8*$^{Hap2}$ in rice protoplasts. (*n* = 3 technical replicates). The values in **f**–**h** are shown as mean ± SEM. Significance is evaluated by the two-sided Student's *t*-test, and *P* values are indicated. **i** Relative expression levels of *OsMYB8* in lodicules of ZH11, *OsMYB8*$^{TF}$/*Osmyb8*$^{ZH}$ transgenic lines and the *Osmyb8*$^{ZH}$ mutant. Values are mean ± SEM. (*n* = 3 biological replicates). Letters above the bars indicate significant differences (*P* < 0.05), as evaluated by one-way ANOVA with Tukey's multiple comparisons test. **j** Comparison of panicles in ZH11, *OsMYB8*$^{TF}$/*Osmyb8*$^{ZH}$ transgenic lines and the *Osmyb8*$^{ZH}$ mutant at 11:30 am in October 2022 in Guangzhou. Scale bars, 1 cm. **k** Number of opened florets in ZH11, *OsMYB8*$^{TF}$/*Osmyb8*$^{ZH}$ transgenic lines, and the *Osmyb8*$^{ZH}$ mutant at different time points of the day in October 2022 in Guangzhou. Values are mean ± SEM. (*n* = 10 panicles). Source data are provided as a Source Data file.

### Potential breeding utilization of *OsMYB8^Hap1* in DFOT improvement of *japonica* varieties

The asynchronous DFOT of *indica* and *japonica* could severely reduce the efficiency of cross-pollination and large-scale hybrid seed production, causing high price of the hybrid seeds. To evaluate the breeding potential of the *OsMYB8^Hap1* allele in DFOT improvement of *japonica* varieties, we introduced the TFB allele of *OsMYB8* into ZH11 through backcrossing and marker-assisted selection. The near-isogenic line (NIL) NIL^TFB with the introgressed *OsMYB8^TFB* allele in ZH11 background was selected from BC₄F₃ plants. We also acquired a chromosome segment substitution line CSSL^9311 that carries a 13 Mb genome segment containing *OsMYB8* from the donor parent 9311 (an *indica* variety) in the *japonica* variety XiuShui134 (XS134) background[29]. Under field conditions, NIL^TFB and CSSL^9311 plants exhibited about 0.5 h earlier DFOT than their recurrent parents (Fig. 7a, b, d, e), without obvious impacts on the tiller number, plant height, heading date, panicle traits, and seed setting rate (Supplementary Fig. 15). In addition, RT-qPCR results showed that the expression levels of *OsMYB8* and *OsJAR1* in NIL^TFB and CSSL^9311 lodicules were higher than that in ZH11 and XS134, respectively (Fig. 7c, f). Therefore, the *OsMYB8^Hap1* allele could promote floret opening in *japonica* varieties, conferring great potential to improve *japonica* DFOT and hybrid seed production.

## Discussion

*Indica* and *japonica* are two subspecies of Asian cultivated rice domesticated from the wild rice *O. rufipogon*[28]. Due to long-term adaptation to different ecological niches, *indica* and *japonica* rice have evolved to exhibit a differentiated DFOT, which might be beneficial for promoting prezygotic reproductive isolation by preventing mating and hybridization, thus facilitating divergence of these two subspecies. However, the asynchronized DFOT of the *indica* and *japonica* parental lines hindered the utilization of the strong heterosis of *indica-japonica* hybrid rice. DFOT is a complex quantitative trait and is easily affected by environmental factors (such as light, temperature and humidity)[30–33]. Although a number of quantitative trait loci (QTL) for DFOT has been previously mapped to different chromosomes of rice, none of them has been molecularly cloned, largely due to the difficulty in precise phenotyping and the small additive effects of the individual locus (typically less than 10% phenotypic variance)[34–38]. In this study, we identified *OsMYB8* as a key regulator of DFOT in both *indica* and *japonica* rice through comprehensive comparative, time-course transcriptome analyses of the lodicules of a representative *indica* TFB and a representative *japonica* ZH11. We showed that expression of *OsMYB8* is up-regulated in the lodicules before floret opening and reaches a plateau at the peak opening time, then gradually declines (Fig. 2d, Supplementary Fig. 5). We further identified *OsJAR1* as a direct target gene of *OsMYB8*, which acts to promote the conversion of JA into biologically active JA-IIe and regulate the expression of genes related to cell osmolality and cell wall remodeling in the lodicules, thus promoting floret opening. Strikingly, we demonstrated that natural variation (6 SNPs) in the 2-kb promoter region of *OsMYB8* confers higher expression of *OsMYB8*, and thus higher expression of *OsJAR1* and higher accumulation of JA-IIe in lodicule cells, ultimately leading to earlier DFOT in *indica* rice compared to *japonica* rice (Fig. 7g). Thus, our results provided insights into the genetic and molecular regulation of DFOT in rice.

Previous studies have documented ample evidence that jasmonate is a vital hormone regulating multiple reproductive processes including flower opening time, and stamen/female organ development, and fertility[39,40]. MYB transcription factors, especially R2R3-MYB members, have been shown to function as key regulators in JA-mediated flower opening and floral organ development in different plant species[41,42]. For example, AtMYB21/24, two homologs of OsMYB8,

are targets of JAZ repressors and could interact with MYC2/3/4/5 to regulate petal elongation (thus leading to petal opening), stamen development, and pollen fertility in *Arabidopsis*[43,44]. The tomato homolog SlMYB21 regulates floret opening as well as carpel and ovule development, through promoting JA biosynthesis in a positive-feedback manner[45]. Previous studies have also reported that the rice mutants deficient in JA biosynthesis and signaling, such as *allene oxide cyclase* (*Osaoc*), *oxophytodienoate reductase 7* (*Osopr7*), *jasmonate resistant 1* (*Osjar1*) and *coronatine insensitive* (*Oscoi1a, Oscoi1b* and *Oscoi2*) mutants, usually exhibit defects in spikelet morphology, anther dehiscence, flower opening and spikelet fertility[46–49]. In this study, we showed that *OsMYB8* acts upstream of *OsJAR1* to promote JA synthesis, and the *Osmyb8* mutant exhibits delayed DFOT, but normal spikelet morphology, pollen maturation, and fertility (Supplementary Fig. 7), suggesting that *OsMYB8* mainly functions in controlling the floret opening time, and it represents an elite gene resource for improving DFOT in rice, with minimal negative pleiotropic effect on stamen development. Notably, phylogenetic analysis identified an *OsMYB8-like* gene in the rice genome that shares 45.2% homology with *OsMYB8* (Supplementary Fig. 3, Supplementary Fig. 16a), and it was preferentially expressed in stamen (Supplementary Fig. 16b), hinting a possible role in regulating stamen development. Further studies are required to elucidate the biological function of the *OsMYB8-like* gene in the future.

Earlier studies have shown that the lodicule undergoes dramatic changes in multiple physiological processes during floret opening and closure, including an increase in soluble sugar content, which leads to a rise in cell osmotic pressure to drive water uptake by the lodicules[13]. An earlier study has demonstrated a critical role of auxin signaling cascade in regulating floret opening. The authors demonstrated that *OsARF2* and *OsARF18* antagonistically regulate the expression of *OsSUT1* (a sucrose transporter gene) to regulate sucrose transport from the source tissues (vegetative organs) into the sink tissues (reproductive organs) and floret opening[50]. It will be worthy to investigate how the *OsMYB8*-mediated JA signaling pathway cross-talks with the auxin signaling pathway to coordinately regulate DFOT in future studies. In addition, recent studies have demonstrated that *DFOT1/ EMF1* negatively regulate DFOT through modulating the activity of pectin methylesterases (PMEs), thus loosening the stiffness of cell walls of lodicules cells[11,20]. In this study, our transcriptome and RT-qPCR analyses indicated that *OsMYB8* regulates the expression levels of various genes related to the cell wall modification. In addition, we also observed that the expression of these genes was altered in the lodicules of the *Osjar1* mutants. Thus, we speculate that *OsMYB8-OsJAR1* module likely regulates floret opening through regulating cell wall relaxation and expansion. It is also worth noting that DFOT is easily affected by environmental factors, such as light, temperature, and humidity[30–33]. We found that the *OsMYB8* promoters of *japonica* and *indica* varieties contain various cis-elements responsive to light, temperature, and hormones based on sequence analysis with PLANTCARE. How these external factors influence the expression of *OsMYB8*, and thus DFOT will be an interesting avenue for future research.

Inter-subspecific *indica-japonica* hybrid rice was proposed to be an important direction of future rice breeding due to its superior heterosis. Nevertheless, current breeding and utilization of *indica-japonica* hybrid rice is still hampered by asynchronized DFOT (and thus low yield of hybrid seed production) in the *indica* and *japonica* parental lines. In this study, we showed that elevating expression level of *OsMYB8* in the *japonica* background or introgression of the Hap1 allele of *OsMYB8* from *indica* varieties into *japonica* varieties could promote DFOT over half an hour, thus effectively reducing the interval between the DFOT of *indica* and *japonica* varieties (Fig. 7a, b, d, e). Importantly, we did not find any significant negative impact of the introgression of *OsMYB8^Hap1* allele on other agronomic

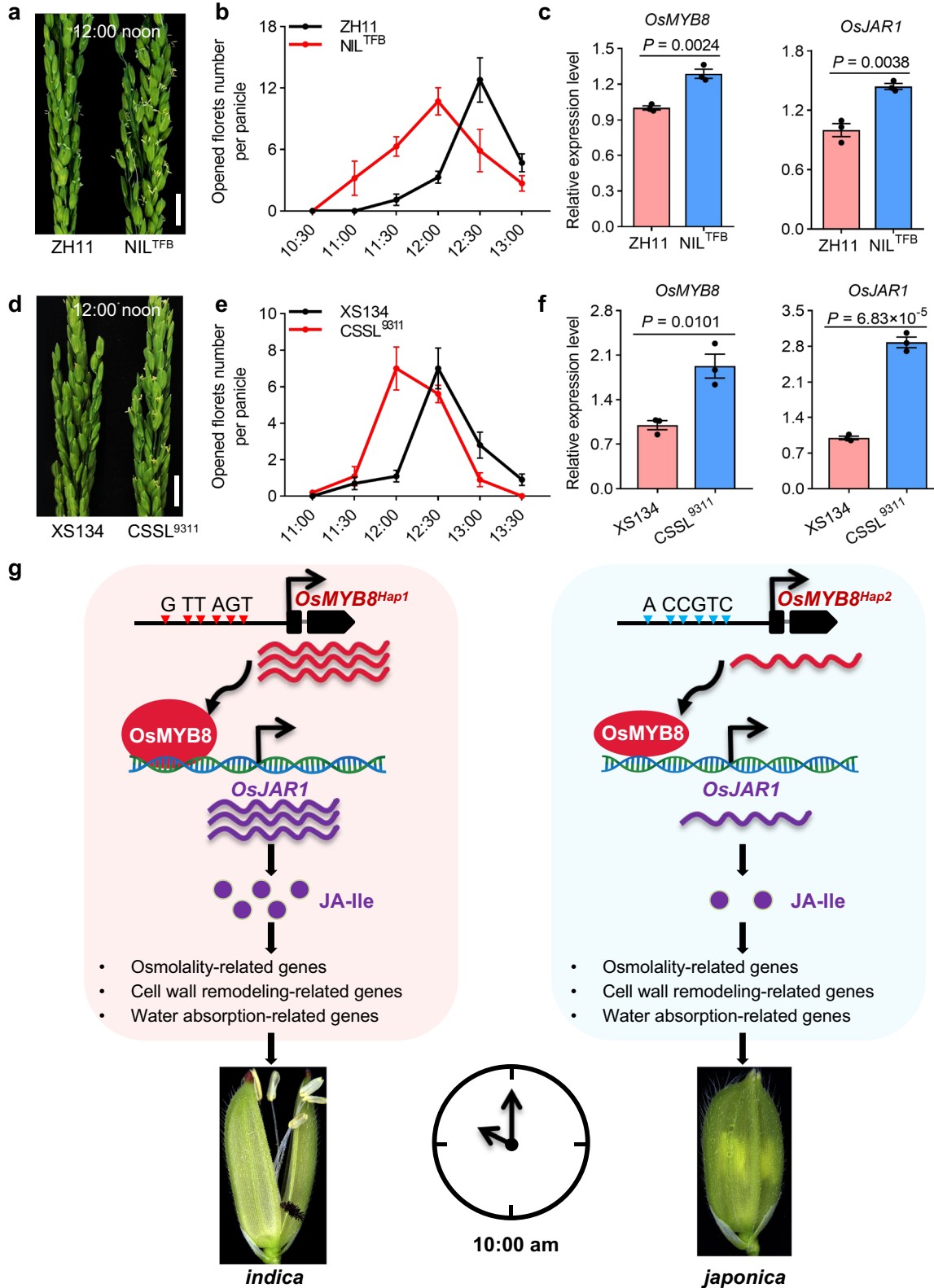

**Fig. 7 | The *indica* allele of *OsMYB8* promotes *japonica* DFOT. a, d** Comparison of panicles in ZH11 and NIL[TFB] (**a**) XS134 and CSSL[9311] (**d**) at 12:00 noon in October 2022 in Guangzhou. Scale bars, 1 cm. **b, e** Number of opened florets in ZH11 and NIL[TFB] (**b**) XS134 and CSSL[9311] (**e**) at different time points of the day in October 2022 in Guangzhou. Values are mean ± SEM. (*n* = 10 panicles). **c, f** Relative expression levels of *OsMYB8* and *OsJAR1* in lodicules of ZH11 and NIL[TFB] (**c**) XS134 and CSSL[9311] (**f**). Values are mean ± SEM. (*n* = 3 biological replicates). Significance is evaluated by the two-sided Student's *t*-test, and *P* values are indicated. **g** A model depicting an *OsMYB8-OsJAR1* module regulating differential DFOT in *indica* and *japonica* rice. Natural variation in the promoter sequences of *OsMYB8* confers higher expression level of *OsMYB8* in *indica*, thus higher accumulation of JA-Ile and earlier DFOT in *indica* as compared to *japonica*. Source data are provided as a Source Data file.

traits (plant architecture and heading date, Supplementary Fig. 14). Thus, further manipulation of *OsMYB8* expression levels may offer a feasible approach to facilitate hybrid seed production and utilization of inter-subspecific *indica-japonica* hybrid rice.

## Methods

### Plant materials and growth conditions

Rice cultivar ZhongHua11 (ZH11) (*Oryza sativa* cv. *Japonica*) and Tian-FengB (TFB) (*Oryza sativa* cv. *Indica*) were used as wild-type controls for mutation or transgenic analyses. The 529 core rice germplasm resources were supplied by National Key Laboratory of Crop Genetic Improvement, Huazhong Agricultural University[26] (Supplementary Data 8). A total of 40 elite rice varieties, including 12 *japonica* varieties and 28 *indica* varieties, were kindly supplied by D. Zhou and were used for DFOT investigation (Supplementary Data 1). XiuShui134 (XS134) and the CSSL line carrying *OsMYB8^{9311}* were kindly provided by B. Hu and C. Chu. For NIL^{TFB} construction, F₁ of ZH11 and TFB plants were backcrossed with ZH11, and NIL^{TFB} were selected in the BC₄F₃ generation with molecular markers. All rice plants were grown in the experimental field of the South China Agricultural University in Guangzhou (23°7′ N, 113°15′ E) from March to November in 2019 to 2023, and in Lingshui, Hainan (18°22′ N, 109°45′ E) from November to April in 2019 to 2023.

### Transgene construct preparation and rice transformation

To generate the *Osmyb8* and *Osjar1* knockout lines, gene-specific guide RNA sequences were designed with an online software toolkit CRISPR-GE[51] and cloned into the CRISPR/Cas9 binary vector *pYLCRISPR/Cas9-Pubi-H*[62]. To generate the *OsMYB8^{TF}*/ZH11 and *OsMYB8^{TF}*/*Osmyb8^{ZH}* transgenic lines, the *OsMYB8* genomic fragment (including 2-kb promoter, coding region, and 1.5 kb 3′UTR) was amplified from TFB gDNA and inserted into the *pCAMBIA1300* vector, forming a recombinant plasmid *p1300-OsMYB8*. To generate the *proOsMYB8::GUS* construct, the 2-kb promoter sequence of *OsMYB8* was amplified from TFB gDNA and cloned into *pCAMBIA1305.1* for plant transformation. To construct the *proOsMYB8::JAR1* plasmid, the *OsJAR1* coding sequence was amplified from ZH11 cDNA and cloned into the *pCAMBIA1300* vector driven by the 2-kb promoter sequence of *OsMYB8* from TFB gDNA. To generate the *proOsMYB8::OsMYB8-GFP* construct, the *OsMYB8* coding sequence (without stop codon) was amplified from ZH11 cDNA and cloned into the *pOx-eGFP* vector driven by the 2-kb promoter sequence of *OsMYB8* from TFB gDNA. The above constructs were transformed into indicated background by Agrobacterium-mediated transformation (Biogle Biological Company, China). Hygromycin (hyg) was used for screening the positive transgenic lines. The mutation sites or gene expression levels were examined by DNA sequencing or RT-qPCR analysis. The primers used for vector constructions are listed in Supplementary Data 11.

### Measurement of the water content in lodicules

100 pairs of lodicules were carefully extracted from the florets and collected as one sample. The fresh weight of each sample was firstly measured using an analytical balance. Then, the samples were dried at 65 °C overnight, and the dry weight of each sample was measured again. The water content of the lodicules was determined by subtracting the dry weight from the fresh weight. Three biological replicates were performed for each sample.

### RNA extraction and RT-qPCR analysis

Total RNAs were extracted from lodicules and other tissues using TRIzol reagent (Thermo Fisher, USA). The complementary DNAs (cDNAs) were synthesized by reverse-transcription according to the manufacturer's instructions (Yeasen, China). RT-qPCR was performed using the LightCycler96 real-time PCR system (Roche, Switzerland) with the qPCR SYBR Green Master Mix (Yeasen, China) according to

the manufacturer's instructions. The *OsActin1* gene (*LOC_Os03g50885*) was used as the internal control. The expression level of genes was calculated using the $2^{-\Delta Ct}$ method. All the primers used for RT-qPCR above are shown in Supplementary Data 11.

### RNA-seq analysis

The lodicules of rice were separated by tweezers and were frozen immediately in liquid nitrogen for RNA-seq analysis. Three biological replicates were performed for each sample, except for the samples of TF with two biological replicates. RNA sequencing was performed at the Azenta company using the Illumina Hiseq platform. For data analysis, the processed reads were compared with Hisat2 v2.0.1[53] to the reference genome MSU Release 7.0[54]. HTSeq v0.6.1[55] was used for quantitative analysis of genes. DESeq2[56] was used for differential expression analysis, and *P*-value < 0.05, absolute log₂FC ≥ 1 were used for screening of the differentially expressed genes. The Gene Ontology (GO) analysis was conducted with the David database (https://david.ncifcrf.gov/home).

### GUS staining and histological observation

For histochemical analysis, the florets about 1–2 h before opening from *proOsMYB8::GUS* transgenic lines were collected for staining using the GUS stain Kit (Coolaber, China) according to the manufacturer's protocol. After staining, the florets were decolorized with 70% (v/v) ethanol and photographed under a Zeiss dissecting microscope.

### Subcellular localization

To determine the subcellular localization of OsMYB8, the coding sequence of *OsMYB8* without the stop codon was amplified from cDNA of the ZH11 and cloned into the *pCAMBIA1305-35S::GFP* vector, resulting in the *35S::OsMYB8-GFP* construct. The nuclear-localized protein D53 fused with mCherry was used as a nuclear marker[57]. The *35S::OsMYB8-GFP* and *p35S::D53-mCherry* fusion constructs were transiently co-transformed into rice protoplasts. The GFP and mCherry fluorescence in protoplasts were observed with a confocal microscope (Zeiss LSM780, Germany).

### Droplet digital PCR

Droplet digital PCR (ddPCR) was performed to select single copy transgene of *OsMYB8^{TF}* in the *Osmyb8^{ZH}* background[58]. The ddPCR reaction mixture consists of 10 μL of 2 × ddPCR super mix for probes (Bio-Rad, USA), 50-100 ng DNA, 900 nM *OsMYB8* primers/250 nM probe (5′ FAM, 3′ BHQ1), 900 nM *OsActin1* primers/250 nM probe (5′ HEX, 3′ BHQ1) and variable ddH₂O in a final volume of 20 μL. The entire reaction mixture was loaded into a disposable plastic cartridge (Bio-Rad, USA) together with 60 μL of droplet generation oil (Bio-Rad, USA) and placed into the droplet generator (Bio-Rad, USA). After processing, the droplets generated from each sample were transferred to a 96-well PCR plate (Eppendorf, Germany). PCR amplification was carried out on a T100 Touch thermal cycler (Bio-Rad, USA) using a thermal profile beginning at 95 °C for 10 min, followed by 45 cycles of 94 °C for 10 s, and 58 °C for 60 s, and ending of 98 °C for 10 min at a ramp rate of 2 °C/s. After PCR, the plate was loaded on the droplet reader (Bio-Rad, USA). The Data were analyzed using the QuantaSoft Analysis Pro software (Bio-Rad, USA).

### DAP-seq analysis

DAP-seq (DNA affinity purification sequencing) is a method used to identify DNA-binding sites of transcription factors[59]. Briefly, 10 μg of ZH11 genomic DNA was broken into 200-bp fragments, fragmented gDNA were constructed into libraries using the VAHTS Universal Pro DNA Library Prep Kit for Illumina (Vazyme, China). The coding sequence of *OsMYB8* was cloned into the *pGEX-4T1* vector to generate the *GST-OsMYB8* recombinant construct. GST-

OsMYB8 recombinant protein or GST protein were purified from *E.coli strain* (DE3) with Glutathinone Sepharose Beads (Sangon, China). The production of GST-OsMYB8 recombinant protein was induced by the addition of 0.4 mM isopropyl b-D-thiogalactopyranoside and grown at 16 °C overnight, and then were purified with GST 4FF Sefinose (TM) Resin Kit (Sangon Biotech, China) according to the manufacturer's protocol. The purified protein was incubated with 500 ng adaptor-ligated gDNA library at room temperature for 2 h before washing away the unbound DNA fragments. The GST proteins were used as the controls. The eluted DNAs were then sequenced using the Illumina HiSeq sequencing platform with two technical replicates.

The clean reads were aligned to the reference genome MSU Release 7.0 using Bowtie2 v2.3.5.1[60]. Aligned reads were sorted and duplicated reads were removed by SAMtools v1.12[61]. Peaks were called using MACS2 v2.1.0[62] ($P < 0.01$). Peaks were identified using the MEME-ChIP online software (https://meme-suite.org/meme/). Putative genes associated with peaks were annotated using ChIP-seeker v1.32.1[63].

### Yeast two-hybrid assay

For Y2H assay, the full-length or truncated CDS of *OsMYB8* were amplified and cloned into the vector *pGBKT7*, and then the recombinant plasmid and *pGADT7* were co-transformed into AH109 yeast cells mediated by PEG4000. The positive transformants were first selected on the (SD)/-Leu/-Trp medium and then transferred to (SD)/-His/-Leu/-Trp and (SD)/-Ade/-His/-Leu/-Trp selection medium (Coolaber, China). The empty vector *pGBKT7* and *pGADT7* were co-transformed as negative controls.

### Electrophoretic mobility shift assay

Native and mutated probes were synthesized and labeled with biotin using the electrophoretic mobility shift assay (EMSA) Probe Biotin Labeling Kit (Beyotime, China). EMSAs were carried out using a Chemiluminescent EMSA Kit (Beyotime, China). Briefly, biotin-labeled probes were incubated for 20 min with the GST or GST-OsMYB8 protein in the binding buffer at room temperature. For competition reaction, 5× and 20× unlabeled cold probes were mixed with the labeled probes. The DNA-protein complex was separated by 5% native polyacrylamide gel electrophoresis and the signal of biotin was photographed using the Biostep Celvin S420 system (Biostep, German). The probes used in this study are listed in Supplementary Data 11.

### Transient transcription dual-LUC assay

For dual-LUC assay, the -2-kb *OsJAR1* promoter was amplified and inserted into the *pGreenII0800-LUC* vector to generate the reporter plasmid. The *OsMYB8* coding sequence without stop codon was amplified and inserted into the *pUC19* vector to generate the effector plasmid. The effector and reporter plasmids were co-transformed into rice protoplasts and incubated in darkness at 28 °C for 12 h. The protoplasts were collected and disintegrated in passive lysis buffer provided in Dual-Luciferase Reporter Gene Assay Kit (Yeasen, 11402ES60). Luciferase activity was also measured using GloMax2020 (Promega) following the manufacturer's instructions. Renilla luciferase (REN) driven by 35S promoter in *pGreenII0800-LUC* was used as the internal control. The relative firefly luciferase activity was counted as the ratio of LUC/REN for each sample.

### Chromatin immunoprecipitation assay

For ChIP assays, rice florets of the ZH11 and transgenic *pOsMYB8::OsMYB8-GFP* plants were harvested and cross-linked in a fixation buffer with 1% (v/v) formaldehyde under vacuum for 20 min. Glycine was added to terminate the cross-linking reaction. The prepared chromatin complexes were sonicated into 200–500 bp fragments and then precleared with protein A magnetic beads (Merck Millipore, USA)[64]. For immunoprecipitations, Anti-GFP (Nanobody) Magnetic beads (ABclonal, China) were added into samples and incubated overnight at 4 °C. DNA was precipitated with magnetic beads. For the real-time-qPCR reaction, the precipitated DNA was recovered and dissolved in water as the template. The enrichment was standardized to the input DNA to obtain the fold enrichment. An unrelated DNA sequence from the rice *OsActin1* gene was used as an internal control. All relevant primers used in the ChIP assay are listed in Supplementary Data 11.

### Measurement of JA-Ile concentration

The quantification of JA-Ile was conducted by Wuhan Metware Biotechnology Co., Ltd, located in Wuhan, China. Approximately 50 mg of lodicules sample was frozen in liquid nitrogen and ground into a fine powder. The sample extracts were analyzed using an LC-ESI-MS/MS system (HPLC: Shim-pack UFLC SHI-MADZU CBM30A system; Shimadzu MS, Applied Biosystems 6500 Triple Quadru-pole). The analytical conditions of HPLC were as follows, LC: column, Waters ACQUITY UPLC HSS T3 C18 (100 mm × 2.1 mm, 1.8 μm); solvent system, water with 0.04% acetic acid (A), acetonitrile with 0.04% acetic acid (B); gradient program, started at 5% B (0–1 min), increased to 95% B (1–8 min), 95% B (8–9 min), finally ramped back to 5% B (9.1–12 min); flow rate, 0.35 mL/min; temperature, 40 °C; injection volume: 2 μL. The ESI source operation parameters were as follows: ion source, ESI±; source temperature 550 °C; ion spray voltage (IS) 5.5 kV (positive), −4.5 kV (negative); curtain gas (CUR) was set at 35 psi, respectively. The contents of JA-Ile were determined using an internal standard method, with three biological replications performed for each sample.

### Measurement of sugar concentration

The quantification of sugar in the lodicules was conducted by Wuhan Metware Biotechnology Co., Ltd, Wuhan, China. The freeze-dried materials were crushed using a mixer mill (MM 400, Retsch) with a zirconia bead for 1.5 min at 30 Hz. 20 mg of powder was diluted to 500 μL with methanol: isopropanol: water (3:3:2 V/V/V), vortexed for 3 min and ultrasound for 30 min. The extract was centrifuged at 13,188 × g under 4 °C for 3 min. 50 μL of the supernatant was mixed with 20 μL internal standard (ribitol, 100 μg/mL) and evaporated under nitrogen gas stream. The evaporated sample was transferred to the lyophilizer for freeze-drying. The residue was used for further derivatization. The derivatization method was as follows: the sample was mixed with 100 μL solution of methoxyamine hydrochloride in pyridine (15 mg/mL). The mixture was incubated at 37 °C for 2 h. Then 100 μL of BSTFA was added into the mixture and kept at 37 °C for 30 min after vortex-mixing. The mixture was analyzed using GC-MS after being diluted to an appropriate concentration. Agilent 7890B gas chromatograph coupled to a 7000 D mass spectrometer with a DB-5MS column (30 m length × 0.25 mm i.d. × 0.25 μm film thickness, J&W Scientific, USA) was employed for GC-MS analysis of sugars. Helium was used as carrier gas, at a flow rate of 1 mL/min. Injections were made in the split mode with a split ratio 3:1 and the injection volume was 3 μL. The oven temperature was held at 170 °C for 2 min, and then raised to 240 °C at 10 °C/min, raised to 280 °C at 5 °C/min, raised to 310 °C at 25 °C/min, and held at the temperature for 4 min. All samples were analyzed in selective ion monitoring mode. The ion source and transfer line temperature were 230 °C and 240 °C, respectively. Three biological replicates were performed for each sample.

### Phylogenetic analysis of R2R3-MYB transcription factors

There are 89 and 124 R2R3-MYB transcription factors in rice and *Arabidopsis* respectively[65]. The full-length protein sequences of R2R3-MYB

transcription factors from both the rice and *Aarabidopsis* genomes were obtained from Ensembl database (http://plants.ensembl.org/index.html). These sequences were then aligned using ClustalW in MEGA7[66]. A neighbor-joining phylogenetic tree was constructed based on the alignment, using the poisson correction method and pairwise deletion of gaps. The reliability of the tree was assessed by bootstrap analysis with 1000 replicates.

## Haplotype analysis

The 2-kb promoter sequence of *OsMYB8* in 3513 cultivated rice accessions were retrieved from the RiceVarMap2.0 database (http://ricevarmap.ncpgr.cn/v2/), and haplotype analysis was carried out using the method in the database. Only haplotypes found in ≥ 10 rice accessions were recorded.

## Nucleotide diversity and fixation index calculation

The raw sequencing data of the 386 cultivated rice were downloaded from NCBI with BioProject PRJNA171289 (>2.5× per genome)[26]. The raw sequencing data of the 185 wild rice were downloaded from NCBI with BioProject accession number PRJNA658215 (>5× per genome)[27]. We mapped the reads to the reference genome MSU Release 7.0 using BWA v0.7.12[67], sorted and indexed the resulting BAM files using SAMtools v1.12[61]. The SNPs were identified using GATK v4.2.0.0[68], the GVCFs of each sample were generated with HaplotypeCaller. We then used CombineGVCFs and GenotypeGVCFs to generate VCF file, the variations were further filtered by VariantFiltration. Nucleotide diversity (π) and fixation index ($F_{ST}$) of *OsMYB8* in wild rice and cultivated rice were calculated using VCFtools v0.1.16[69] with 20-kb windows and 2-kb steps on the 100-kb region.

## Statistical analysis

We used GraphPad Prism 8 for statistical analysis. Data are mean ± SEM. Two-sided unpaired Student's *t*-test was used to test the significant difference between two groups. Three groups and more were analyzed by one-way analysis of variance (ANOVA) with Tukey's multiple comparisons test.

## Reporting summary

Further information on research design is available in the Nature Portfolio Reporting Summary linked to this article.

## Data availability

DAP-seq and RNA-seq data generated in this study have been deposited in the NCBI Sequence Read Archive database under the accession number PRJNA1000954 and PRJNA1000956, respectively. Source data are provided with this paper.

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

## Acknowledgements

We thank B. Hu and C. Chu (South China Agricultural University, SCAU) for providing Xiushui134 (XS134) and the CSSL line carrying *OsMYB8^{9311}*. This work was supported by the National Natural Science Foundation of China (No. 31921004 to H.H.; No. 31991222 to H.H.; No. 32172056 to R.S.), Hainan Yazhou Bay Seed Laboratory (No. B23YQ1515 to R.S.; No. B23CQ15FP to R.S.), the Natural Science Foundation of Guangdong Province-Guangzhou City Collaborative Key Project (No. 2019B1515120061 to R.S.) and the Double First-class Discipline Promotion Project (No. 2021B10564001 to R.S.).

## Author contributions

H.W. and R.S. conceived and designed experiments. Y.G. performed most of the experiments, W.D., Y.H. and M.Z. analyzed the data. C.X. and Q.T. characterized the genotypes and phenotypes of the edited lines. Y.L., Y.F. and R.Y. designed the CRISPR target and constructed the plasmid library. D.Z., X.L. and X.Z. carried out the phenotype investigation of transgenic rice and other rice germplasms. Y.G., Y.H., H.W. and R.S. wrote the manuscript. H.W. revised the article.

## Competing interests

The authors declare no competing interests.
