## [Peer Review File · Nature Communications]

Natural variation in OsMYB8 confers diurnal floret opening time divergence between indica and japonica subspeciesReviewers' Comments:

Reviewer #1:

Remarks to the Author:

The asynchronized DFOT of the indica and japonica parental lines hindered the utilization of the strong heterosis of indica-japonica hybrid rice. Therefore, identifying the primary QTLs that regulate days to flowering could yield valuable genetic loci facilitating the production of inter-subspecific indica-japonica hybrid seeds.

This paper presents a comparison of the DFOT divergence in indica and japonica rice subspecies. The authors further conducted an in-depth investigation into the regulatory role of OsMYB8 in DFOT.

Notably, the presence of the Hap1 sequence in OsMYB8 is associated with an earlier floret opening time in indica rice varieties. Additionally, isogenic lines with the OsMYB8Hap1 genotype demonstrated the ability to enhance floret opening time in japonica varieties. The study also reveals that OsMYB8 directly governs the expression of OsJAR1, leading to the promotion of jasmonic acid (JA) accumulation during floret opening. These findings hold implications for rice breeding strategies. Nonetheless, certain supplementary experiments remain necessary to achieve comprehensive conclusions before proceeding with publication.

Main questions:

1. Could the authors provide insights into the spatial-temporal expression pattern of OsMYB8 over the course of a day? Additionally, does the elevated expression level of OsMYB8 contribute to an earlier floret opening time in indica varieties? I highly recommend that the authors undertake a comparative analysis of the spatial-temporal expression levels of OsMYB8 during floret opening in both TFB and ZH11.
2. The endogenous concentration of jasmonates closely correlates with floret opening time. How does the spatial-temporal concentration of JA change in TFB, ZH11 and OsMYB8TF/ZH11 over the course of a day? Is there a correlation between the distribution of jasmonates and the expression level of OsMYB8? Is there a correlation between the content of jasmonic acid (JA) and the size of the lodicules during the process of floret opening?
3. Given that OsMYB8 is a plausible ortholog of AtMYB23 and AtMYB24, both of which have been documented to engage with JAZ and MYC proteins in the context of stamen development regulation, I am intrigued by the potential correlation between impaired flower opening time and atypical stamen development. If such a connection exists, it would be prudent to at least address the implications of stamen functions in the context of DFOT. What is the expression pattern of OsMYB8 in the stamen during anthesis?
4. Building upon the previous inquiry, is there a parallel alteration in the expression pattern of OsJAR1, mirroring that of OsMYB8, across the diurnal cycle within the lodicules of TFB and ZH11, respectively?
5. What is the expression level of OsJAZs during floret opening? Is the expression of OsJAR1 still inducible by exogenous JA application in the Osmyb8 mutant?
6. In comparison to the wild type, the DFOT of osmyb8 occurs approximately 0.5 to 1 hour later. Could you provide an explanation for the fact that the florets of the osmyb8 mutant still open around noon? Moreover, does the application of exogenous JA have the potential to restore the aberrant DFOT observed in Osmyb8? Additionally, has there been a change in the distribution pattern of JA within the Osmyb8 mutant during the process of floret opening?
7. Can the dissimilar days to floret opening time (DFOT) between indica and japonica variants be nullified through the exogenous application of JA? Is this effect contingent on the presence of OsMYB8?

Minor questions:

1. The genomes of the transgenic plants OsMYB8TF/ZH11#1 and OsMYB8TF/ZH11#2 harbored two copies of OsMYB8. Hence, it is imperative to create a transgenic plant bearing OsMYB8TF in the osmyb8/ZH11 genetic background. Alternatively, the authors should consider generating an OsMYB8ZH11/ZH11 line as a control. Presently, the available data do not allow for the conclusion that the promoter variation of OsMYB8 is responsible for promoting an earlier floret opening time, either

through an independent manner or in a dosage-dependent fashion.

2. Can the OsMYB8-GFP construct restore the floret flowering time of Osmyb8 in the backgrounds of both TFB and ZH11?
3. The expression of OsAmy2, Os4BGlu, OsSWEET11/15, OsEXPB7, OsXTH16, OsPG17, and OsNIP1;1 was altered in the Osmyb8 and Osjar1 mutants. Nevertheless, there is currently no direct evidence demonstrating that these genes regulate lodicule hydration and expansion.
4. Is the expression level of OsJAR1 consistently higher in OsMYB8Hap1/Osmyb8ZH than in the Osmyb8ZH mutant? Likewise, does a similar question arise regarding the content of JA in these two transgenic lines?
5. The peak opening time of ZH11 florets exhibited variation in Figs. 5, 6, and 7, ranging from 11 AM to 12:30 PM. I have concerns regarding the findings presented in Fig. 5i. When comparing the peak opening time of the OsJARcom transgenic lines, why does the maximum flowering time correspond to 11 AM, as opposed to the 12 or 12:30 PM of other control ZH11 instances?

Reviewer #2:

Remarks to the Author:

This manuscript describes the function of the OsMYB8 rice transcription factor on the diurnal timing of flower opening. The data include targeted mutation of key genes as well as transcriptomic and promoter-binding studies, and make a strong case for the role of OsMYB8 in rice flower opening. They find that OsMYB8 promotes expression of the OsJAR1 gene, whose product makes the hormone jasmonyl-Ile. Both factors are needed to promote flower opening in the morning, and genetic epistasis analyses also support the proposed pathway. Moreover, the timing of flower opening differs between different rice varieties, and this trait can be modified by moving OsMYB8 alleles (with different promoter haplotypes) between varieties. The work may therefore lead to improvements in rice breeding through generating indica and japonica varieties whose flowers open at the same time of day and are therefore easier to cross with each other. I have a few specific suggestions for improvement:

From Figure S3, it appears that there is also another OsMYB paralog in the same group with OsMYB8 and the two Arabidopsis MYBs that mediate jasmonate responses in flowers. It would be nice to show the expression pattern of that other rice MYB - is it also expressed in flowers, and might it act redundantly with OsMYB8?

In the Discussion, it would be nice to compare the role of jasmonate and the MYB factors that mediate jasmonate response between rice and dicots such as Arabidopsis and tomato (see Schubert et al., *The Plant Cell*, and references cited therein). In Arabidopsis, jasmonate induces MYB21, which in turn promotes jasmonate responses. Is there a similar positive feedback in rice also? Do other jasmonate pathway genes (other than OsJAR1) also depend on OsMYB8? Does jasmonate have any effect on stamen or style development in rice, as in Arabidopsis or tomato?

Figure 3 and Figure 6 each use an indica OsMYB8 gene in a japonica background. In one case it is called OsMYB(TF) (Figure 3, in a wild-type japonica), in the other case it is called OsMYB(Hap1) (Figure 6, in a myb8 mutant in the japonica background). Are these in fact the same transgene? If so, they could be given the same name.

I am wondering whether there is a link of flower opening time to circadian rhythms - are there circadian rhythm transcriptomic datasets available in rice, and if so, does the circadian rhythm regulate MYB8 or the jasmonate pathway?

Is there information on the diurnal flowering time of the Hap3 variants?

line 420, check the sentence for a possible missing word.

Reviewer #3:

Remarks to the Author:

The manuscript "Natural variation in OsMYB8 confers diurnal floret opening time divergence between indica and japonica subspecies" by Gou and colleagues identifies the transcription factor OsMYB8 as a regulator of diurnal floret opening time (DFOT) in rice. It controls the transcription of OsJAR1 encoding for the key enzyme for bioactivation of the plant hormone jasmonic acid (JA), which conjugates JA to the amino acid to the amino acid isoleucine. JA-Ile seems to be one of the factors determining DFOT. Because of differential regulation of the transcription factor, flowering times in indica and japonica cultivars differ significantly causing non-matching flowering-opening times between these subspecies. The authors furthermore show that the variation in the regulation of OsMYB8 caused by modifications in the promoter sequences can be exploited in order to utilize naturally occurring alleles in breeding programs for homogenization of flowering times between the two subspecies.

The manuscript is really interesting to read, scientifically sound and of high quality.

However, the discussion of the results is a bit poor, mainly highlighting on the possible application. I recommend to elaborate more scientifically interesting points. One such point can be the link between OsMYB8 and jasmonate. The authors show that flowering time is under control of jasmonate. However, jar1 mutant of rice display also an open husk phenotype (<https://onlinelibrary.wiley.com/doi/10.1111/j.1365-3040.2008.01790.x>), and rice allene oxide cyclase mutants also show flower morphology phenotypes (<https://onlinelibrary.wiley.com/doi/10.1111/tpj.12115>), and the rice coi2 (a jasmonate receptor mutant) can't open its anthers properly (<https://academic.oup.com/pcp/article/64/4/405/6874509>). These phenotypes are not described for the osmyb8 mutants investigated here, hence rice utilizes JA at different steps during flower development, including heading time, flower morphology, anther dehiscence, flower opening etc. OsMYB8 seems to be the key for flower opening, but not for other steps (including closure of flowers). From my point of view this is not really explained sufficiently in the discussion. I even suggest to extend the results in order to show the regulation patterns of other JA-related genes in addition to OsJAR1, e.g. which JAZ genes and which catabolic genes are differentially regulated between WT and osmyb8.

Furthermore OsJAR1 has a close homologue, OsJAR2

(<https://onlinelibrary.wiley.com/doi/10.1111/pce.12201>,

<https://www.sciencedirect.com/science/article/pii/S0006291X11008229?pes=vor>), which is able to conjugate JA and isoleucine. Does OsJAR2 contain the TTHGGY-motive in its promoter or not?

These are some ideas and references which might help in improving the discussion.

Some details:

1. Line 191-194: The type of mutation and impact on amino acid sequence should be shown more clearly, may be in an additional supplemental figure, as this is very important for the evaluation.
2. Line 244-245: A list of genes in cell wall modification and JA pathway is highly recommended to show. These can be points to elaborate in the discussion.
3. Line 285: There is no Fig. 5g. Please check Figure plate 5.
4. Line 372-373: You should explain in one or two sentences why this would be an advantage. As yield seems not be altered readers might wonder what the advantage of improved DFOT would be. Just pick up the thread about hybrid rice production from the beginning and it becomes clear.
5. Figure 3 c, f, k: You better label the y-axis in the figures here and in later figures similarly: opened florets per panicle.

Minor points:

1. Line 124, Figure 1: Adjust y-axis in d and e to the same scale?
2. Line 124, Figure 1: Probably a more common abbreviation for standard error of the mean is either SE or SEM (in capital letters, respectively). Please clarify, and correct accordingly throughout the manuscript.
3. Line 124, Figure 1: change y-axis to mg instead of g; indicate more clearly whether this is weight per lodicule, water content per lodicule?

4. Line 156, Supplementary Fig. 2: Abbreviations used in the figure need a better explanation. What means MF and BP? Readers have to guess. Suggestion: make a legend where you explain all the abbreviations such as T9, TF etc., and use it in each figure legend where it is used. It will help the reader to get the point faster.
5. Line 180: Correct: write either 'leaf' or 'leaves'
6. Line 210, Fig. 4a: better label upper row as free-GFP, lower row as OsMYB8-GFP. The current labelling causes confusion.

Reviewer #4:

Remarks to the Author:

The manuscript "Natural variation in OsMYB8 confers diurnal floret opening time divergence between indica and japonica subspecies" investigates the mechanism that regulates daily flower (floret) opening and closure in rice. This topic is of note as the natural variation of locus for transcription factor OsMYB8 showed potential of synchronous floret opening important for more efficient harvesting of hybrid rice seeds. Through well-designed sampling and global gene expression analysis, the authors identified the transcription factor MYB8 as a responsive factor for synchrony. Lodicule is the mechanical organ driving the movement. In lodicules, MYB8 expression profiles are consistent with the occurrence of the movement. Molecular analyses to probe binding sites isolated the target JA-Ile promoter motifs. Genetic approach making these CRISPER/Cas9 induced mutant lines determined the role of inducer for the target genes. Further authors found that the indica MYB8 locus has dominant positive promoter activity and demonstrated the application in japonica to establish the strong synchrony.

Overall, this is an outstanding manuscript contributing to the research field. The experiments in this manuscript are well thought out and organized. Previously a DFOT factor, DUF642, is determined, which negatively regulates rice DFOT through lodicule cell-wall modulation, in this report authors demonstrated another genetic DFOT phenotype (variation) due to promoter haplotypes. The finding and application are outstanding noteworthy points in plant and the breeding sciences. However, there are a few details that would be added to the text to solve difficulties in finding the words and means for general readers in the journal.

Major notes:

Strong points: Consistency of the context. Seamless in experimental results from lodicule expression to functional analysis among Haplotype of promoter, MYB8 binding to JAR1 promoter and CRISPER/Cas9 gene-targeting mutation.

Weak points:

Fig. 5a shows 2 electropherograms as DNA sequencing of Crispr/cas9 derivatives. The Osjar1#2 sequence looks like 2 base pair deletion. But it might be one base pair deletion. It should be sequenced from the lower side with an anti-directional oligonucleotide primer.

Fig. 6a shows haplotypes of OsMYB8 promoter. These OsMYB8 promoters determine the strength of transcription. Although authors determined 3 variations in the promoter, readers would expect other related regulatory motifs, e.g., for lodicule specific expression and light response. Please describe in discussion or supplementary figure.

Minor notes:

Line 95 OsMYB8 is italic now. Not Italic.

Line 159 The T9 time point were,, was?

Line 239 log₂FC, log₂FC (folds change).

Line 265 to regulates, regulate.

Line 298 Petin is Pectin.

Line 317 (TrJ) is italic? What name is referred to Aus?

Line 580 What is PLB?

Line 727 32. Tsutomu, I. et al. ,, Ishimaru, T. et al.

Supplementary Fig. 9 Line 114 indicate is indicates.

Responses to reviewers' comments

Reviewers' comments:

Reviewer #1 (Remarks to the Author):

The asynchronized DFOT of the indica and japonica parental lines hindered the utilization of the strong heterosis of indica-japonica hybrid rice. Therefore, identifying the primary QTLs that regulate days to flowering could yield valuable genetic loci facilitating the production of inter-subspecific indica-japonica hybrid seeds.

This paper presents a comparison of the DFOT divergence in indica and japonica rice subspecies. The authors further conducted an in-depth investigation into the regulatory role of OsMYB8 in DFOT. Notably, the presence of the Hap1 sequence in OsMYB8 is associated with an earlier floret opening time in indica rice varieties. Additionally, isogenic lines with the OsMYB8^{Hap1} genotype demonstrated the ability to enhance floret opening time in japonica varieties. The study also reveals that OsMYB8 directly governs the expression of OsJAR1, leading to the promotion of jasmonic acid (JA) accumulation during floret opening. These findings hold implications for rice breeding strategies. Nonetheless, certain supplementary experiments remain necessary to achieve comprehensive conclusions before proceeding with publication.

Response: We are very appreciated for the reviewer's careful reviewing and comments of our manuscript.

Main questions:

1. Could the authors provide insights into the spatial-temporal expression pattern of OsMYB8 over the course of a day? Additionally, does the elevated expression level of OsMYB8 contribute to an earlier floret opening time in indica varieties? I highly recommend that the authors undertake a comparative analysis of the spatial-temporal expression levels of OsMYB8 during floret opening in both TFB and ZH11.

Response: Thank you for the valuable suggestions. To address these questions, we conducted RT-qPCR analysis to compare the expression level of *OsMYB8* in the lodicules of TFB and ZH11 at various time points, including 18:00 the day before floret opening and

multiple time points on the day of floret opening (8:00 am, 9:00 am, 10:00 am/peak floret opening time for TFB, 11:00 am and 12:00 noon/peak floret opening time for ZH11). The results showed that the expression level of *OsMYB8* was lower at the day before floret opening, and then gradually increased towards opening time, and decreased thereafter. Additionally, we found that the expression level of *OsMYB8* in *indica* lodicules was significantly higher than that in *japonica* before floret opening. These observations suggest that the elevated expression level and an earlier peak expression time of *OsMYB8* in TFB, in comparison to ZH11, likely contribute to the earlier floret opening time in *indica* varieties. The result was shown in new Supplementary Fig. 5 and was described in line 183-193.

Supplementary Fig. 5: The expression level of *OsMYB8* in the lodicules of TFB and ZH11 at different time points. RT-qPCR analysis showing the expression level of *OsMYB8* in the lodicules of TFB and ZH11 at 18:00 the day before floret opening and different time points on the day of floret opening, including 8:00 am, 9:00 am, 10:00 am (peak floret opening time for TFB, TF), 11:00 am and 12:00 noon (peak floret opening time for ZH11, ZF). Values are mean \pm SEM. (n=2 biological replicates). Significance is determined by two-sided Student's *t*-test, ** $P < 0.01$.

2. The endogenous concentration of jasmonates closely correlates with floret opening time. How does the spatial-temporal concentration of JA change in TFB, ZH11 and *OsMYB8*TF/ZH11 over the course of a day? Is there a correlation between the distribution of jasmonates and the expression level of *OsMYB8*? Is there a correlation between the

content of jasmonic acid (JA) and the size of the lodicules during the process of floret opening?

Response: We appreciate the reviewer's comments. He *et al*¹ measured the JA content in florets of Wuyunjing 7 (a *japonica* cultivar) by HPLC-electrospray ionization tandem mass spectrometry system (HPLC-MS/MS) during natural anthesis period from 7:00 am to 17:00. The JA levels in the rice florets were relatively low and constant (10-15 ng g⁻¹ DW) from 2-5 h before floret opening (7:00 am-10:00 am), there was a slight increase in JA levels at 1 h before opening (25 ng g⁻¹ DW), and it rapidly reached a peak during anthesis (12:00 noon, 103 ng g⁻¹ DW). After the floret closing (14:00-17:00), the JA levels subsequently decreased. The result indicated that endogenous JA is involved in the regulation of floret opening in rice.

We like to point out that lodicule is a tiny organ lying within the base of florets, and the concentration of JA is very low in lodicules at the time before floret opening (Fig. 5d, e; Fig. 6f). Due to the tiny size of the lodicules, obtaining sufficient quantities of lodicule samples for JA measurement over the time course and in a number of genetic backgrounds for metabolome analysis is almost a mission impossible. More than 500 pairs of lodicules are needed for each biological replicate. Besides, only several florets in a plant are at a proper stage during floret opening in one day, and the diurnal floret opening time is easily affected by environmental factors, therefore, it is of great difficulty to collect enough samples of lodicules at different time points to perform the measurement of JA concentration over the course of a day, especially at night, when collecting large number of proper floret is impossible in the field. Therefore, in response to this comment, we switched to investigate the dynamic change of endogenous JA by utilizing the transgenic rice carrying a JA perception biosensor OsJAZ6-VENUS containing the JA sensor module (Jas-VENUS) and the red fluorescent protein mCherry, which could sensitively visualize JA content *in vivo*². We showed that the VENUS signals were obviously decreased from 18:00 (the day before floret opening) to 9:00 am and 10:00 am, then vanished at 11:00 am and 12:00 noon (the day of floret opening). Consistently, a time-course quantification of VENUS fluorescence normalized to mCherry signals further confirmed the gradually decreased VENUS fluorescence signals. These results indicated that endogenous JA had accumulated as early

as at 9:00 am, then continued to increase at 10:00 am and 11:00 am and reached its maximum at 12:00 noon. This accumulation pattern is in high accordance with the expression profiles of *OsMYB8* (new supplementary Fig. 5) and the size changes of the lodicules during the process of floret opening (Fig.1h, Response Fig. 1c). The result was shown in the Response Fig. 1.

Response Fig. 1 The VENUS fluorescence of ZH11 lodicule cells at different time points. **a** The VENUS signals of ZH11 lodicule cells at 18:00 at the day before floret opening and different time points on the day of floret opening, including 8:00 am, 9:00 am, 10:00 am, 11:00 am and 12:00 noon (peak floret opening time for ZH11). Scale bars, 10 μ m. **b** The quantification of VENUS fluorescence normalized to mCherry signals at different time points. Values are mean \pm SEM. (n = 6). **c** The lodicule volume of ZH11 at different time points. Values are mean \pm SEM. (n = 10).

3. Given that *OsMYB8* is a plausible ortholog of *AtMYB23* and *AtMYB24*, both of which have been documented to engage with JAZ and MYC proteins in the context of stamen development regulation, I am intrigued by the potential correlation between impaired

flower opening time and atypical stamen development. If such a connection exists, it would be prudent to at least address the implications of stamen functions in the context of DFOT. What is the expression pattern of *OsMYB8* in the stamen during anthesis?

Response: We appreciate the reviewer's valuable suggestions. As suggested, we performed expression analysis of *OsMYB8* in the ZH11 and TFB stamens during anthesis. RT-qPCR results showed a gradual increase expression level of *OsMYB8* in the stamens of both ZH11 and TFB as the opening time approaching. The result was shown in the Response Fig. 2.

Response Fig. 2 The expression level of *OsMYB8* in the TFB and ZH11 stamens at different time points. RT-qPCR analysis showing the expression level of *OsMYB8* in the TFB and ZH11 stamens at 18:00 the day before floret opening and different time points on the day of floret opening, including 8:00 am, 9:00 am, 10:00 am (peak floret opening time for TFB, TF), 11:00 am and 12:00 noon (peak floret opening time for ZH11, ZF). Values are mean \pm SEM. (n=3 biological replicates). Significance is determined by two-sided Student's *t*-test, * $P < 0.05$, ** $P < 0.01$.

Next, we examined the effect of *OsMYB8* on stamen and pollen development. In contrast to the role of *AtMYB21* and *AtMYB24* in the regulation of stamen development, we showed that both *OsMYB8* knockout lines in ZH11 and TFB backgrounds exhibited normal spikelet and stamen development, and the anthers of *Osmyb8* mutants also dehiscenced normally (new Supplementary Fig.7 a-c). Besides, we found that most pollen grains from the mutants could be stained by a 1% I₂-KI (potassium iodide) solution, and the pollen fertility of mutants were comparable to that of WT plants (new Supplementary Fig.7 d-f).

These results demonstrated that *OsMYB8* may primarily function in floret opening but confer minimal influence on the anther and pollen development, indicating a functional divergent of these *MYB* homologs between rice and *Arabidopsis*. This new result was shown in new Supplementary Fig.7 and was discussed in line 212-219.

Supplementary Fig. 7 *OsMYB8* did not affect rice fertility and anther dehiscence.

a, b The florets of ZH11, *Osmyb8^{ZH}*, TFB and *Osmyb8^{TF}* before opening (**a**) and after opening (**b**). Scale bars, 1 mm. **c** The stamens of ZH11, *Osmyb8^{ZH}*, TFB and *Osmyb8^{TF}* before opening. Scale bars, 1 mm. **d** Images of the pollens of ZH11, *Osmyb8^{ZH}*, TFB and *Osmyb8^{TF}* stained with 1% I₂-KI (potassium iodide) solution. Scale bars, 50 μm. **e, f** The pollen fertility of ZH11 and *Osmyb8^{ZH}* (**e**), TFB and *Osmyb8^{TF}* (**f**) based on I₂-KI staining. Values are means ± SEM. (n = 10). Significance is evaluated by the two-sided Student's *t*-test.

4. Building upon the previous inquiry, is there a parallel alteration in the expression pattern of *OsJAR1*, mirroring that of *OsMYB8*, across the diurnal cycle within the lodicules of TFB and ZH11, respectively?

Response: As suggested, we analyzed the expression level of *OsJAR1* in the lodicules of TFB and ZH11 at 18:00 the day before floret opening and multiple time points on the day

of floret opening (8:00 am, 9:00 am, 10:00 am/peak floret opening time for TFB, 11:00 am and 12:00 noon/peak floret opening time for ZH11). The result showed that the expression level of *OsJAR1* increases as the floret opening time approaches (new Supplementary Fig.9d), similar as that of *OsMYB8* in lodicules before floret opening. The result was shown in the new Supplementary Fig.9d and was discussed in line 262-264.

Supplementary Fig. 9 OsMYB8 promote the expression of *OsJAR1*. **a** The FPKM of *OsJAR1* in the *Osmyb8^{TF}* and *Osmyb8^{ZH}* lodicule transcriptome. Data are shown as means ± SEM. (n = 3 biological replicates). Significance is evaluated by the two-sided Student's *t*-test, and *P* values are indicated. **b, c** RT-qPCR analysis of relative *OsJAR1* expression level in lodicules of the *Osmyb8* mutants (**b**) and *OsMYB8^{TF}*/ZH11 lines (**c**). Data are shown as means ± SEM. (n = 3 biological replicates). Significance is evaluated by the two-sided Student's *t*-test, and *P* values are indicated. **d** The expression level of *OsJAR1* in the lodicules of TFB and ZH11 at 18:00 the day before floret opening and different time points on the day of floret opening, including 8:00 am, 9:00 am, 10:00 am (peak floret opening time for TFB, TF), 11:00 am and 12:00 noon (peak floret opening time for ZH11, ZF). Values are mean ± SEM. (n = 2 biological replicates). Significance is determined by two-sided Student's *t*-test, ***P* < 0.01.

5. What is the expression level of OsJAZs during floret opening? Is the expression of OsJAR1 still inducible by exogenous JA application in the *Osmyb8* mutant?

Response: As suggested, we reexamined the transcriptomic data of lodicules in TFB and ZH11, and analyzed the expression levels of all 15 *OsJAZs* genes³ in the lodicules during floret opening. As shown in the heatmap of the average FPKM values of *OsJAZs*, a large proportion of the *OsJAZs* exhibited elevated expression and reached high-level of expression at the peak floret opening time (Response Fig. 3), probably due to the feedback activation of *OsMYC2* as described in previous studies^{4,5}. This result further suggests the activation of JA signaling in the lodicules during floret opening.

94.86	119.13	148.79	106.83	118.09	116.87	98.14	OsJAZ1 (LOC_Os04g55920)
0.39	1.89	89.01	0.78	2.28	0.86	8.97	OsJAZ2 (LOC_Os07g05830)
6.25	13.36	6.38	4.99	9.09	9.90	8.29	OsJAZ3 (LOC_Os08g33160)
648.18	2267.33	363.62	454.55	1416.63	1369.92	908.77	OsJAZ4 (LOC_Os09g23660)
0.02	0.26	4.64	0.01	0.06	0.28	1.59	OsJAZ5 (LOC_Os04g32480)
71.16	521.38	2102.23	72.00	394.31	437.23	1700.62	OsJAZ6 (LOC_Os03g28940)
72.02	341.39	950.91	48.75	200.16	302.06	628.35	OsJAZ7 (LOC_Os07g42370)
4.84	14.46	107.77	1.92	3.89	12.10	107.44	OsJAZ8 (LOC_Os09g26780)
4.87	413.23	860.11	3.66	131.36	191.03	1121.29	OsJAZ9 (LOC_Os03g08310)
3.04	260.61	1143.21	2.76	111.57	114.52	1266.13	OsJAZ10 (LOC_Os03g08330)
12.29	1616.34	5753.39	4.00	461.12	840.30	7057.24	OsJAZ11 (LOC_Os03g08320)
7.08	94.31	483.71	4.33	29.39	41.48	343.59	OsJAZ12 (LOC_Os10g25290)
1.74	1.34	66.27	1.20	0.21	1.95	102.59	OsJAZ13 (LOC_Os10g25230)
0.02	0.05	0.21	0.02	0.00	0.00	0.06	OsJAZ14 (LOC_Os10g25250)
2.71	13.56	0.97	16.80	7.53	8.82	1.22	OsJAZ15 (LOC_Os03g27900)
T18	T9	TF	Z18	Z9	Z11	ZF	

Response Fig. 3: The expression heatmap of *OsJAZs* in lodicules transcriptome of TFB and ZH11. The numbers in the heatmap represent the average FPKM values.

To assess the effect of exogenous JA application on the *Osmyb8* mutant, we exogenously applied 2 mM MeJA to ZH11 and *Osmyb8^{ZH}* mutants at 10:00 am (about 2 h before peak floret opening in ZH11). After 0.5 h, both ZH11 and *Osmyb8^{ZH}* mutants treated with MeJA exhibited peak floret opening at the same time, indicating that exogenous MeJA application could restore the delayed DFOT phenotype of *Osmyb8* to the wild type (Response Fig. 4 a-c). This result further demonstrates that *OsMYB8* is involved in JA

synthesis but not in JA signaling. Additionally, RT-qPCR analysis showed that exogenous MeJA application could significantly elevated the expression level of *OsJAR1* in ZH11 and *Osmyb8^{ZH}* mutants (Response Fig. 4 d), suggesting that the JA-induced expression of *OsJAR1* expression is independent of *OsMYB8*.

Response Fig. 4: Exogenous MeJA treatment of ZH11 and *Osmyb8^{ZH}* mutants. a, b The panicles of ZH11 and *Osmyb8^{ZH}* after treatments with solutions supplemented without MeJA (a) or with 2 mM MeJA (b) for 30 min, respectively. Scale bars, 1 cm. **c** The opened florets number of ZH11 and *Osmyb8^{ZH}* after treatments with solutions supplemented with 2 mM MeJA or without MeJA for 30 min. Values are means \pm SEM. (n = 6). **d** The expression level of *OsJAR1* in the lodicules of ZH11 and *Osmyb8^{ZH}* after treatments with solutions supplemented with 2 mM MeJA or without MeJA for 30 min. Values are means \pm SEM. (n = 3 technical replicates). Significance is evaluated by the two-sided Student's *t*-test.

6. In comparison to the wild type, the DFOT of *Osmyb8* occurs approximately 0.5 to 1 hour later. Could you provide an explanation for the fact that the florets of the *Osmyb8* mutant

still open around noon? Moreover, does the application of exogenous JA have the potential to restore the aberrant DFOT observed in *Osmyb8*? Additionally, has there been a change in the distribution pattern of JA within the *Osmyb8* mutant during the process of floret opening?

Response: We appreciate the reviewer's comments. As shown in the new Supplementary Fig. 9a, the expression level of *OsJARI* in *Osmyb8^{ZH}* is about one-fourth of that in ZH11, and is about half of that in TFB. This partial down-regulation of *OsJARI* in these mutants would not completely block JA biosynthesis, so these mutants could still exhibit peak floret opening at a delayed time compared to their WT plants. If the *OsJARI* is completely knocked out in rice, it would display a scattered floret opening phenotype (random floret opening throughout the day) without peaking opening time (Fig 5c).

Moreover, as shown in the Response Fig. 4, exogenous MeJA application could restore the delayed DFOT phenotype of *Osmyb8* to the wild type, indicating the delayed DFOT phenotype of *Osmyb8* is due to its insufficient JA content in lodicules (see also our response to Comment 5).

As the time point of 1-2 h before opening is regarded as a crucial stage for floret opening^{6,7}. Moreover, in this study, we showed that *OsMYB8* exhibited peak expression level at the time of 1-2 h before floret opening. Therefore, we just measured the JA-Ile content in the lodicules of ZH11 and *Osmyb8^{ZH}*, and the result showed that the JA-Ile content in the lodicules of the *Osmyb8^{ZH}* mutant was significantly lower than that of ZH11 (Fig. 5d). Together with other molecular and genetic evidence presented in the manuscript (Fig. 4f-h, Fig. 5d-k), we concluded that *OsMYB8* regulates JA-Ile content through regulating *OsJARI* in lodicules to promote floret opening in rice. Actually, as mentioned above (see our response to Comment 2), it is technically impossible to measure the JA concentration in lodicules at different time points during the course of a day.

7. Can the dissimilar days to floret opening time (DFOT) between indica and japonica variants be nullified through the exogenous application of JA? Is this effect contingent on the presence of *OsMYB8*?

Response: Thank you for the suggestions. In our study, we refer to rice DFOT (diurnal

floret opening time) specifically as the time when the flowers open within the day, which is different from the heading date. JA could effectively promote the DFOT in both *indica* and *japonica* rice varieties. Yan *et al*⁸ showed that exogenous MeJA application could promote DFOT of different *indica* and *japonica* varieties in the field, with *indica* (0.04 mmol/L) being more sensitive to MeJA treatment than *japonica* (0.4 mmol/L). Therefore, application of MeJA with proper concentration on the *japonica* rice at a proper time could promote its DFOT and might abolish the different DFOT between *indica* and *japonica*. However, as for commercial production of *indica-japonica* hybrid seeds in the field, plants of the female parent (usually a male sterile *japonica* line) are planted in rows side-by-side with plants of the male parent (usually an *indica* rice). It is technically impossible to specifically spray MeJA on the *japonica* parent without affecting the *indica* parent, and exogenous MeJA spray in the field would amplify instead of nullifying the difference of DFOT in the *indica* and *japonica* parents. Therefore, the genetic improvement of rice DFOT may offer the most cost-effective and feasible strategy to nullify asynchronized DFOT of *indica* and *japonica*.

As our response to Comment 5 mentioned, when exogenously treated with 2 mM MeJA, both ZH11 and *Osmyb8^{ZH}* exhibited early DFOT (see the Response Fig. 4a-d), indicating that JA-induced floret opening in rice is not contingent on the presence of *OsMYB8*.

Minor questions:

1. The genomes of the transgenic plants *OsMYB8TF/ZH11#1* and *OsMYB8TF/ZH11#2* harbored two copies of *OsMYB8*. Hence, it is imperative to create a transgenic plant bearing *OsMYB8TF* in the *osmyb8/ZH11* genetic background. Alternatively, the authors should consider generating an *OsMYB8ZH11/ZH11* line as a control. Presently, the available data do not allow for the conclusion that the promoter variation of *OsMYB8* is responsible for promoting an earlier floret opening time, either through an independent manner or in a dosage-dependent fashion.

Response: Thank you for the suggestions. In our original manuscript, we have already generated the transgenic plant bearing the *OsMYB8^{TF}* haplotype (*OsMYB8^{Hap1}*) in the *Osmyb8/ZH11* genetic background (*OsMYB8^{Hap1}/Osmyb8^{ZH}*, Fig.6). RT-qPCR analysis

showed that the expression levels of *OsMYB8* in lodicules of *OsMYB8^{Hap1}/Osmyb8^{ZH}* plants were about 3-fold that of ZH11 (Fig. 6i), similar to the expression difference of *OsMYB8* between TFB and ZH11 (Fig. 2d, e). Phenotypic analysis showed that the *OsMYB8^{Hap1}/Osmyb8^{ZH}* plants exhibited about 0.5 and 1 h earlier DFOT than ZH11 and the *Osmyb8^{ZH}* mutant, respectively (Fig. 6j, k). These results together indicate that the natural variation in the *OsMYB8* promoter between ZH11 and TFB contributes to the differential expression of *OsMYB8*, and thus divergence of DFOT in these two subspecies. To make a more explicit description of this transgenic line, we renamed *OsMYB8^{Hap1}/Osmyb8^{ZH}* as *OsMYB8^{TF}/Osmyb8^{ZH}* in this revised manuscript.

2. Can the *OsMYB8*-GFP construct restore the floret flowering time of *Osmyb8* in the backgrounds of both TFB and ZH11?

Response: Thank you for the suggestions. In this study, we generated two transgenic lines with stable expression of *OsMYB8-GFP* driven by the native *OsMYB8* promoter from TFB in ZH11 background, which was used to examine the binding capacity of *OsMYB8* to the *OsJAR1* promoter *in vivo* (Supplementary Fig. 10a). We also demonstrated that the *OsMYB8-GFP* transgene exhibits higher *OsMYB8* expression level than ZH11, and could be translated into the stable *OsMYB8-GFP* fusion protein (Supplementary Fig. 10b-c). In this revised manuscript, we further add the new result showing that the *OsMYB8-GFP* lines exhibited about 1 h earlier DFOT than ZH11 as expected (Supplementary Fig. 10d-e). All these lines of evidence strongly support that the *OsMYB8-GFP* transgene is normally expressed and could function in DFOT regulation, thus should be suitable for performing ChIP-qPCR assays.

We have not generated *OsMYB8-GFP* transgenic lines in TFB background in this study.

Supplementary Fig. 10: Generation and verification of *OsMYB8-GFP* plants. **a** Schematic diagram of the vector structure used for constructing *OsMYB8-GFP* materials. The *pOsMYB8^{TF}* means the promoter was amplified from TFB. **b** The relative expression level of *OsMYB8* in the lodicules of ZH11 and *OsMYB8-GFP* materials. Values are mean \pm SEM. (n = 3 biological replicates). Significance is evaluated by the two-sided Student's *t*-test, and *P* values are indicated. **c** Detection of *OsMYB8-GFP* by immunoblotting with anti-GFP antibody. ZH11 was included as a negative control, and anti-Actin was included for immunoblotting to show similar loadings. Total proteins were extracted from florets of ZH11 and the *OsMYB8-GFP* homozygous plants at 1-3 h before opening. **d** Comparison of panicles in ZH11 and *OsMYB8-GFP* at 10:30 am in June 2022 in Guangzhou. Scale bars, 1 cm. **e**, The number of opened florets per panicle in ZH11 and *OsMYB8-GFP* at different time points of the day. Values are means \pm SEM. (n = 8).

3. The expression of *OsAmy2*, *Os4BGlu*, *OsSWEET11/15*, *OsEXPB7*, *OsXTH16*, *OsPG17*, and *OsNIP1;1* was altered in the *Osmyb8* and *Osjar1* mutants. Nevertheless, there is currently no direct evidence demonstrating that these genes regulate lodicule hydration and expansion.

Response: Thanks for the comment. Precious studies showed that the floret opening is attributed to the alterations of cell turgor and osmotic pressure in lodicules⁹⁻¹¹. Consistently, in our study, we found that a series of genes associated with cell osmolality and cell wall remodeling altered their expression in the *Osmyb8* and *Osjar1* mutants. These genes belong to several gene super-families, such as sugar metabolism (*OsAmy2*, *Os4BGlu* and others), sugar transporter (*OsSWEET11/15*), cell wall organization (*OsEXPBs*, *OsPMEs*, *OsXTHs*, etc.). These findings prompt us to propose that these genes might work together to regulate lodicule hydration and expansion. It is notable that the gene families mentioned above contain many members in rice genome, for example, there are 10 α -amylases (including *OsAmy2*)¹², 40 β -glucosidases (including *Os4BGlu*)¹³, 21 sugar transporters (including *OsSWEET11/15*)¹⁴, 18 β -expansins (including *OsEXPB7*)¹⁵, 29 xyloglucan endotransglucosylases (including *OsXTH16*)¹⁶, 44 polygalacturonases (including *OsPG17*)¹⁷ and 33 aquaporins (including *OsNIP1;1*)¹⁸ in rice genome, respectively. Considering that the possible redundancy within these gene families, it is plausible that knocking out a single gene or a few gene members might not affect DFOT phenotype. To make a more proper conclusion, we revised the description in lines 324-326 in the original text with “These results suggest that *OsJAR1* likely influence floret opening by modulating the expression of genes related to lodicule hydration and expansion.”

4. Is the expression level of *OsJAR1* consistently higher in *OsMYB8Hap1/Osmyb8ZH* than in the *Osmyb8ZH* mutant? Likewise, does a similar question arise regarding the content of JA in these two transgenic lines?

Response: Yes, we have further analyzed the *OsJAR1* expression in ZH11, *OsMYB8^{TF}/Osmyb8^{ZH}* and *Osmyb8^{ZH}*. The results showed that the expression level of *OsJAR1* was consistently higher in *OsMYB8^{TF}/Osmyb8^{ZH}* than in ZH11 and the *Osmyb8^{ZH}* mutant (new supplementary Fig. 13 a). Furthermore, the JA-Ile content in the lodicules of *OsMYB8^{TF}/Osmyb8^{ZH}* was significantly higher than that of ZH11 (new supplementary Fig. 13 b), indicating that *OsMYB8* regulates floret opening through promoting the expression of *OsJAR1* in lodicules. We revised the description in line 371-374 in the original text with “Consistently, the expression level of *OsJAR1* was higher in *OsMYB8^{TF}/Osmyb8^{ZH}* than in

ZH11 and the *Osmyb8^{ZH}* mutant, and the JA-Ile content in the lodicules of *OsMYB8^{TF}/Osmyb8^{ZH}* was significantly higher than that of ZH11 (Supplementary Fig. 13).”

Supplementary Fig. 13 *OsMYB8^{TF}* elevates *OsJAR1* expression level and JA-Ile content in *Osmyb8^{ZH}* mutant. **a** The relative expression level of *OsJAR1* in lodicules of ZH11, *OsMYB8^{TF}/Osmyb8^{ZH}* and *Osmyb8^{ZH}* mutant. Values are mean \pm SEM. (n=3 biological replicates). Letters above the bars indicate significant differences ($P < 0.05$), as evaluated by one-way ANOVA with Tukey’s post-hoc analysis. **b** The JA-Ile content in lodicules at 9:00 am of ZH11 and *OsMYB8^{TF}/Osmyb8^{ZH}*. Values are mean \pm SEM. (n=3 biological replicates). Significance is evaluated by the two-sided Student’s *t*-test at each time point, and *P* values are indicated.

5. The peak opening time of ZH11 florets exhibited variation in Figs. 5, 6, and 7, ranging from 11 AM to 12:30 PM. I have concerns regarding the findings presented in Fig. 5i. When comparing the peak opening time of the *OsJARcom* transgenic lines, why does the maximum flowering time correspond to 11 AM, as opposed to the 12 or 12:30 PM of other control ZH11 instances?

Response: Thanks for the comments. As rice DFOT is easily affected by the environmental factors, which generally result in the DFOT variability of a rice variety between different planting seasons. In the early season in Guangzhou (March-June), ZH11 heads in June and its florets usually open between 11:00 am-12:00 noon (Fig 1d), while in the late season (August-November), ZH11 heads in October and its florets usually open between 12:00 noon-13:00 (Fig 1e). The DFOT phenotypes shown in Fig. 3 c, f, k and Fig. 5 c&i were

investigated in June 2021 and June 2022, respectively, while those presented in Fig. 5k, Fig. 6k, and Fig. 7b&e were examined in October 2022. We have indicated the months for the DFOT investigation in the corresponding Figure Legends.

Reviewer #2 (Remarks to the Author):

This manuscript describes the function of the OsMYB8 rice transcription factor on the diurnal timing of flower opening. The data include targeted mutation of key genes as well as transcriptomic and promoter-binding studies, and make a strong case for the role of OsMYB8 in rice flower opening. They find that OsMYB8 promotes expression of the OsJAR1 gene, whose product makes the hormone jasmonyl-Ile. Both factors are needed to promote flower opening in the morning, and genetic epistasis analyses also support the proposed pathway. Moreover, the timing of flower opening differs between different rice varieties, and this trait can be modified by moving OsMYB8 alleles (with different promoter haplotypes) between varieties. The work may therefore lead to improvements in rice breeding through generating indica and japonica varieties whose flowers open at the same time of day and are therefore easier to cross with each other.

Response: We are very appreciated for the reviewer's careful reviewing and positive comments of our manuscript.

I have a few specific suggestions for improvement:

From Figure S3, it appears that there is also another OsMYB paralog in the same group with OsMYB8 and the two Arabidopsis MYBs that mediate jasmonate responses in flowers. It would be nice to show the expression pattern of that other rice MYB - is it also expressed in flowers, and might it act redundantly with OsMYB8?

Response: Thanks for the suggestion. As the reviewer suggested, we have detected the transcript levels of the *OsMYB8-like* gene in various tissues, including root, stem, leaf, panicle and flower organs (lemma, palea, stamen, pistil and lodicule). In contrast to *OsMYB8*, *OsMYB8-like* gene was highly expressed in stamen but not in lodicule (Response Fig. 5). Additionally, we conducted an analysis of the haplotype of *OsMYB8-like* using Ricevarmap2 (http://ricevarmap.ncpgr.cn/hap_net/) and found no obvious differentiation

between *indica* and *japonica* varieties. Therefore, our research focused on *OsMYB8*.

Response Fig. 5: The expression pattern analysis of *OsMYB8-like*. Relative expression levels of *OsMYB8-like* in various tissues of ZH11. Values are mean ± SEM. (n=3 biological replicates).

In the Discussion, it would be nice to compare the role of jasmonate and the MYB factors that mediate jasmonate response between rice and dicots such as *Arabidopsis* and tomato (see Schubert et al., *The Plant Cell*, and references cited therein). In *Arabidopsis*, jasmonate induces MYB21, which in turn promotes jasmonate responses. Is there a similar positive feedback in rice also? Do other jasmonate pathway genes (other than *OsJAR1*) also depend on *OsMYB8*? Does jasmonate have any effect on stamen or style development in rice, as in *Arabidopsis* or tomato?

Response: We appreciate the reviewer’s insightful comments. Based on the suggestion, we compared the previous findings on the role of jasmonate and the MYB factors in *Arabidopsis*, rice and tomato, and added the following paragraph in discussion part:

“Previous studies have documented ample evidence that jasmonate is a vital hormone regulating multiple reproductive processes including flower opening time, and stamen/female organ development and fertility^{19,20}. MYB transcription factors, especially R2R3-MYB members, have been shown to function as key regulators in JA-mediated flower opening and floral organ development in different plant species^{21,22}. For example, *AtMYB21/24*, two homologs of *OsMYB8*, are targets of JAZ repressors and could interact with *MYC2/3/4/5* to regulate petal elongation (thus leading to petal opening), stamen

development and pollen fertility in *Arabidopsis*^{23,24}. The tomato homolog SIMYB21 regulate floret opening as well as carpel and ovule development, through promoting JA biosynthesis in a positive-feedback manner²⁵. Notably, we showed in this study, that *OsMYB8* acts upstream of *OsJAR1* to promote JA synthesis, and primarily acts to promote floret opening without obviously affecting the development of other floret organs such as stamen and style (Supplementary Fig. 7), suggesting that the functions of these homologous MYB transcription factors may have diverged in monocotyledonous and dicotyledonous plant species. Previous studies have also reported that the rice mutants deficient in JA biosynthesis and signaling, such as *allene oxide cyclase (Osaoc)*, *oxophytodienoate reductase 7 (Osopr7)*, *JASMONATE RESISTANT 1 (Osjar1)* and *coronatine insensitive mutants (Oscoi1a, Oscoi1b and Oscoi2)*, usually exhibit defects in spikelet morphology, anther dehiscence, flower opening and spikelet fertility²⁶⁻²⁹. Here we show that the *Osmyb8* mutant exhibits normal spikelet morphology, pollen maturation and fertility (Supplementary Fig. 7), suggesting that *OsMYB8* mainly functions in controlling the floret opening time, and it represents an elite gene resource for improving DFOT in rice, with minimal negative pleiotropic effect on stamen development.”

Regarding the comment “In *Arabidopsis*, jasmonate induces MYB21, which in turn promotes jasmonate responses. Is there a similar positive feedback in rice also? ”

As suggested, we performed RT-qPCR experiments to investigate whether JA could induce the expression of *OsMYB8*, and the results indicate JA could significantly induce the expression of *OsMYB8*, implying a positive feedback loop regulation of *OsMYB8* on JA biosynthesis. Nevertheless, more detailed analysis of this regulation is beyond the scope of this study.

Response Fig. 6: The expression level of *OsMYB8* in lodicules of ZH11 after 2 mM MeJA treatment. Values are mean ± SEM. (n = 3 biological replicates).

Regarding the comment “Do other jasmonate pathway genes (other than *OsJAR1*) also depend on *OsMYB8*?”

As suggested, we performed overlapping analysis of 72 genes (Response Table 1) potentially involved in JA synthesis (39), signal transduction (25), and metabolism (8) with DEGs in the lodicules transcriptomes of *Osmyb8^{TF}* and *Osmyb8^{ZH}* mutants. Only 4 genes (*OsDAD1-1*, *OsJAR1* and *OsJAZ9/11*) showed differential expression in both *Osmyb8^{TF}* and *Osmyb8^{ZH}* (Response Fig.7), suggesting that *OsMYB8* only specifically regulates a small set of the JA pathway genes.

Response Table 1. The 72 genes related to JA pathway

Gene name	Gene id	Gene name	Gene id	Gene name	Gene id
OsACX1	LOC_Os06g01390	OsLOXL-2	LOC_Os03g52860	OsJAZ10	LOC_Os03g08330
OsACX2	LOC_Os11g39220	OsOPR1	LOC_Os06g11290	OsJAZ11	LOC_Os03g08320
OsACX3	LOC_Os06g24704	OsOPR2	LOC_Os06g11280	OsJAZ12	LOC_Os10g25290
OsAOC	LOC_Os03g32314	OsOPR3	LOC_Os06g11260	OsJAZ13	LOC_Os10g25230
OsAOS1	LOC_Os03g55800	OsOPR4	LOC_Os06g11240	OsJAZ14	LOC_Os10g25250
OsAOS2	LOC_Os03g12500	OsOPR5	LOC_Os06g11210	OsJAZ15	LOC_Os03g27900
OsAOS3	LOC_Os02g12680	OsOPR6	LOC_Os06g11200	OsCOI1a	LOC_Os01g63420
OsAOS4	LOC_Os02g12690	OsOPR7	LOC_Os08g35740	OsCOI1b	LOC_Os05g37690
OsDAD1-1	LOC_Os02g43700	OsOPR8	LOC_Os02g35310	OsCOI2	LOC_Os03g15880
OsDAD1-2	LOC_Os11g04940	OsOPR9	LOC_Os01g27240	OsMYC1	LOC_Os01g13460
OsDAD1-3	LOC_Os08g04800	OsOPR10	LOC_Os01g27230	OsMYC2	LOC_Os10g42430
OsDAD1-4	LOC_Os10g41270	OsJMT1	LOC_Os05g01140	OsMYC3	LOC_Os01g50940

Response Table 1. The 72 genes related to JA pathway (continued)

Gene name	Gene id	Gene name	Gene id	Gene name	Gene id
OsLOX1	LOC_Os02g10120	OsJMT2	LOC_Os06g20920	OsMYC4	LOC_Os01g64560
OsLOX2	LOC_Os03g08220	OsJAR1	LOC_Os05g50890	OsMYC5	LOC_Os02g02820
OsLOX3	LOC_Os03g49260	OsJAR2	LOC_Os01g12160	OsMYC6	LOC_Os04g47040
OsLOX4	LOC_Os03g49350	OsJAZ1	LOC_Os04g55920	OsMYC7	LOC_Os04g47080
OsLOX5	LOC_Os03g49380	OsJAZ2	LOC_Os07g05830	OsCYP94B3a	LOC_Os11g05380
OsLOX6	LOC_Os04g37430	OsJAZ3	LOC_Os08g33160	OsCYP94B3b	LOC_Os05g37250
OsLOX7	LOC_Os05g23880	OsJAZ4	LOC_Os09g23660	OsCYP94B3c	LOC_Os01g63930
OsLOX8	LOC_Os08g39850	OsJAZ5	LOC_Os04g32480	OsCYP94C2b	LOC_Os12g05440
OsLOX9	LOC_Os08g39840	OsJAZ6	LOC_Os03g28940	OsIAR1	LOC_Os01g37960
OsLOX10	LOC_Os11g36719	OsJAZ7	LOC_Os07g42370	OsIAR2	LOC_Os06g47620
OsLOX11	LOC_Os12g37260	OsJAZ8	LOC_Os09g26780	OsJAO1	LOC_Os03g18030
OsLOX12	LOC_Os12g37350	OsJAZ9	LOC_Os03g08310	OsJAO2	LOC_Os01g61610

Response Fig. 7: The overlapping analysis of JA-related genes and the lodicules transcriptome of *Osmyb8* mutants. The numbers in the heatmap represent the average FPKM values.

Regarding the comment “Does jasmonate have any effect on stamen or style development in rice, as in *Arabidopsis* or tomato?”

Previous studies have well documented the role of jasmonates on floret development, including stamen and style in rice. Nevertheless, as shown in the Supplementary Fig. 7 in

the revised manuscript, the normal development of floret organs of *Osmyb8* mutants indicate that *OsMYB8* does not obviously affect stamen or style development (see also our response to Comment 3 of the reviewer #1). We discussed this point in Discussion part (line 428-451) in the revised manuscript.

Figure 3 and Figure 6 each use an indica *OsMYB8* gene in a japonica background. In one case it is called *OsMYB(TF)* (Figure 3, in a wild-type japonica), in the other case it is called *OsMYB(Hap1)* (Figure 6, in a *myb8* mutant in the japonica background). Are these in fact the same transgene? If so, they could be given the same name.

Response: Thanks for the suggestion. *OsMYB8^{TF}* and *OsMYB8^{Hap1}* are indeed the same transgene. We have revised “*OsMYB8^{Hap1}/Osmyb8^{ZH}*” as “*OsMYB8^{TF}/Osmyb8^{ZH}*” in the revised manuscript to avoid confusion.

I am wondering whether there is a link of flower opening time to circadian rhythms - are there circadian rhythm transcriptomic datasets available in rice, and if so, does the circadian rhythm regulate MYB8 or the jasmonate pathway?

Response: We appreciate the reviewer’s insightful comments. Circadian rhythms are endogenous biological oscillations with a period of approximately 24 hours, enabling organisms to anticipate daily environmental changes. In flowering plants, floret opening often exhibit a circadian floral movement rhythm. Previous studies have shown that silencing the core circadian clock genes (*NaLHY* and *NaZTL*) strongly altered floret opening rhythms in different ways in *Nicotiana attenuata*, suggesting that the internal circadian clock could regulate floret opening time³⁰. Given that rice florets also frequently open at specific times of the day under natural conditions, it is tempting to speculate that circadian rhythm could regulate *OsMYB8* and the jasmonate pathway to control the rice DFOT. However, it is of great difficulty to collect enough samples of lodicule (a tiny organ) at different time points to address this point, carefully examine the relationship of circadian clock in regulating *OsMYB8*, JA and DFOT is beyond the scope of this study, which is a very interest aspect to be addressed in our future studies.

Is there information on the diurnal flowering time of the Hap3 variants?

Response: Thanks for the suggestion. As Hap3 was mostly detected in *TrJ* and *Aus* (Fig. 6b), which are two rice subpopulations with a small quantity in several rice germplasm collections. For example, in a recent study conducted by Ge *et al*³¹, 400 rice varieties collected from China were re-sequenced and analyzed, only 3 varieties were classified under the *tropical japonica* (*TrJ*) group, and none of these varieties belonged to the *Aus* or *aromatic* groups. In this study, we focused more on the DFOT phenotype of Hap1, which is mainly *Ind*, and Hap2, mainly *TeJ*.

Considering that the early DFOT of *indica* rice may be beneficial for improving fertility in regions with high-temperature conditions³², and *TrJ* and *Aus* are usually planted in tropical region, we speculate that the DFOT of Hap3 also have an early DFOT similar to Hap1. Consistent with our speculation, in the late season of 2023, we cultivated a *TrJ* variety called “Lemont”, and its DFOT coincided with TFB, exhibiting an earlier DFOT phenotype.

line 420, check the sentence for a possible missing word.

Response: We apologize for the typing error in the previous manuscript, and we have added the missing words “*Indica-japonica* hybrid rice” in the revised manuscript.

Reviewer #3 (Remarks to the Author):

The manuscript “Natural variation in OsMYB8 confers diurnal floret opening time divergence between *indica* and *japonica* subspecies“ by Gou and colleagues identifies the transcription factor OsMYB8 as a regulator of diurnal floret opening time (DFOT) in rice. It controls the transcription of OsJAR1 encoding for the key enzyme for bioactivation of the plant hormone jasmonic acid (JA), which conjugates JA to the amino acid to the amino acid isoleucine. JA-Ile seems to be one of the factors determining DFOT. Because of differential regulation of the transcription factor, flowering times in *indica* and *japonica* cultivars differ significantly causing non-matching flowering-opening times between these subspecies. The authors furthermore show that the variation in the regulation of OsMYB8 caused by modifications in the promoter sequences can be exploited in order to utilize

naturally occurring alleles in breeding programs for homogenization of flowering times between the two subspecies.

The manuscript is really interesting to read, scientifically sound and of high quality.

Response: We appreciate the reviewer's careful review and positive comments of our manuscript.

However, the discussion of the results is a bit poor, mainly highlighting on the possible application. I recommend to elaborate more scientifically interesting points. One such point can be the link between *OsMYB8* and jasmonate. The authors show that flowering time is under control of jasmonate. However, *jar1* mutant of rice display also an open husk phenotype (<https://onlinelibrary.wiley.com/doi/10.1111/j.1365-3040.2008.01790.x>), and rice allene oxide cyclase mutants also show flower morphology phenotypes (<https://onlinelibrary.wiley.com/doi/10.1111/tbj.12115>), and the rice *coi2* (a jasmonate receptor mutant) can't open its anthers properly (<https://academic.oup.com/pcp/article/64/4/405/6874509>). These phenotypes are not described for the *osmyb8* mutants investigated here, hence rice utilizes JA at different steps during flower development, including heading time, flower morphology, anther dehiscence, flower opening etc. *OsMYB8* seems to be the key for flower opening, but not for other steps (including closure of flowers). From my point of view this is not really explained sufficiently in the discussion. I even suggest to extend the results in order to show the regulation patterns of other JA-related genes in addition to *OsJAR1*, e.g. which JAZ genes and which catabolic genes are differentially regulated between WT and *osmyb8*.

Response: We are grateful of the reviewer's insightful comments. We have expanded the discussion about the link between *OsMYB8* and jasmonate (JA) more comprehensively, which is also raised by the reviewer #1. We appreciate the reviewer for providing these reference papers. After carefully reading the reference, we thought that the main difference between *OsMYB8* and genes involved in JA biosynthesis and signaling is that *OsMYB8* primarily functions in controlling floret opening time, as the *Osmyb8* mutants do not display significant alterations in spikelet morphology, anther dehiscence and fertility. Therefore, *OsMYB8* is an elite gene source for improving DFOT in rice. The discussion

have been added to the revised discussion part (line 428-451).

Furthermore OsJAR1 has a close homologue, OsJAR2 (<https://onlinelibrary.wiley.com/doi/10.1111/pce.12201>, <https://www.sciencedirect.com/science/article/pii/S0006291X11008229?pes=vor>) , which is able to conjugate JA and isoleucine. Does OsJAR2 contain the TTHGGY-motive in its promoter or not?

These are some ideas and references which might help in improving the discussion.

According to the reviewer's suggestion, we performed sequence analysis of *OsJAR2* promoter, and found that multiple "TTHGGY" motifs in its promoter. Nevertheless, based on the RNA-seq data of the TFB and ZH11 lodicules at different time points, we found that *OsJAR2* is expressed at very low or undetectable levels in the lodicules, with FPKM values < 1 (Response Table 2). In addition, a previous study has reported that *OsJAR1* is the predominant enzyme catalyzing the formation of JA-Ile in rice floret³³. Therefore, we focused on *OsJAR1* in this study.

Reponse Table 2. The FPKM of *OsJAR2* in lodicules transcriptome.

Sample	FPKM
T18	0.303401127
T9	0.381505611
TF	0.396458333
Z18	0.472217243
Z9	0.218637218
ZF	0.496970178
Z11	0.140729051
TFB	0.014558042
Osmyb8^{TF}	0
ZH11	0.020712108
Osmyb8^{ZH}	0

Some details:

1. Line 191-194: The type of mutation and impact on amino acid sequence should be shown more clearly, may be in an additional supplemental figure, as this is very important for the

evaluation.

Response: Thanks for the suggestion. We have provided amino acid sequences deduced from the DNA sequencing data of the two mutant lines and highlighted the positions of frame-shift and premature stop. The data are presented in Supplemental Figure 6.

OsMYB8	MATRMCGRAGEPAVRKGFWTLEEDLILVSYISQNGEGSWDNLARSAGLNRRNGKSCRLRWLNLYLRPGVRRG	70
Osmyb8 ^{ZH} #1	MATRMCGRAGEPACRAQGFVDAGGGPHPRQLHLAKRRRI LGQPRALCRAEPEEREELQAAVAQLPEARCAAG	70
Osmyb8 ^{ZH} #2	MATRMCGRAGEPARAQGFVDAGGGPHPRQLHLAKRRRI LGQPRALCRAEPEEREELQAAVAQLPEARCAAG	70
Osmyb8 ^{TF} #1	MATRMCGRAGEPASRAQGFVDAGGGPHPRQLHLAKRRRI LGQPRALCRAEPEEREELQAAVAQLPEARCAAG	70
Osmyb8 ^{TF} #2	MATRMCGRAGEPARAQGFVDAGGGPHPRQLHLAKRRRI LGQPRALCRAEPEEREELQAAVAQLPEARCAAG	70
OsMYB8	SITPEEDMVIRELHNRWIKRWSKI AKHLPGRTDNEIKNYWRTKIHRKPRGRSQLLQEFCE D A M G M S T T	140
Osmyb8 ^{ZH} #1	QHHAGGGHGHGPAPLPVGEQVVQDRQAPPRPDRQRDQEELLEDDDTQEAARQEPAAAAGAVRGRHGHHGHVHH	140
Osmyb8 ^{ZH} #2	QHHAGGGHGHGPAPLPVGEQVVQDRQAPPRPDRQRDQEELLEDDDTQEAARQEPAAAAGAVRGRHGHHGHVHH	140
Osmyb8 ^{TF} #1	QHHAGGGHGHGPAPLPVGEQVVQDRQAPPRPDRQRDQEELLEDDDTQEAARQEPAAAAGAVRGRHGHHGHVHH	140
Osmyb8 ^{TF} #2	QHHAGGGHGHGPAPLPVGEQVVQDRQAPPRPDRQRDQEELLEDDDTQEAARQEPAAAAGAVRGRHGHHGHVHH	140
OsMYB8	TSEAASTSAASSGQSQA S EGVWDEYMQASSFPHPPELVSF AADHHEMAGVGEVAAAAAAQFVPT EFGFNDG	210
Osmyb8 ^{ZH} #1	HQRGGVDVGVERPEPGQERRLG.....	162
Osmyb8 ^{ZH} #2	HQRGGVDVGVERPEPGQERRLG.....	162
Osmyb8 ^{TF} #1	HQRGGVDVGVERPEPGQERRLG.....	162
Osmyb8 ^{TF} #2	HQRGGVDVGVERPEPGQERRLG.....	162
OsMYB8	FWNFVDNFWETMPVSDVV	228
Osmyb8 ^{ZH} #1	162
Osmyb8 ^{ZH} #2	162
Osmyb8 ^{TF} #1	162
Osmyb8 ^{TF} #2	162

Supplementary Fig. 6 Alignment of the amino acid sequences of *OsMYB8* in WT and the *Osmyb8* mutants. The predictive amino acid sequences of *OsMYB8* in the *Osmyb8^{ZH}#1*, *Osmyb8^{ZH}#2*, *Osmyb8^{TF}#1* and *Osmyb8^{TF}#2* mutant lines are deduced from their DNA sequencing results. Frame shifts in *Osmyb8* mutants are indicated by the red box. Red triangles indicate the premature terminations in the *Osmyb8* mutants.

2. Line 244-245: A list of genes in cell wall modification and JA pathway is highly recommended to show. These can be points to elaborate in the discussion.

Response: Thanks for the suggestion. We have listed the genes related to the JA pathway and cell wall modification in Supplementary Data 6 as suggested. We have discussed it in the revised manuscript (line 464-469) as follows:

“In this study, our transcriptome and RT-qPCR analyses indicated that *OsMYB8* regulates the expression levels of various genes related to the cell wall modification. In addition, we also observed that the expression of these genes was altered in the lodicules of the *Osjar1* mutants. Thus, we speculate that *OsMYB8*-*OsJAR1* module likely regulates

floret opening through regulating cell wall relaxation and expansion.”

3. Line 285: There is no Fig. 5g. Please check Figure plate 5.

Response: Thanks! Fig. 5g is actually included in the original Figure plate 5, which is close to Fig. 5f in the original version. We have readjusted it to be more distinct in the revised version.

4. Line 372-373: You should explain in one or two sentences why this would be an advantage. As yield seems not be altered readers might wonder what the advantage of improved DFOT would be. Just pick up the thread about hybrid rice production from the beginning and it becomes clear.

Response: Thanks for the suggestion. We have added the following sentence in our revised manuscript line 382-384: “The asynchronous DFOT of *indica* and *japonica* could severely reduce the efficiency of cross-pollination and large-scale hybrid seed production, causing high price of the hybrid seeds”.

5. Figure 3 c, f, k: You better label the y-axis in the figures here and in later figures similarly: opened florets per panicle.

Response: Thanks. We have revised it as suggested.

Minor points:

1. Line 124, Figure 1: Adjust y-axis in d and e to the same scale?

Response: Thank you for pointing it out. The scale bars of y-axis in Figure 1d and 1e have been unified in our revised manuscript.

2. Line 124, Figure 1: Probably a more common abbreviation for standard error of the mean is either SE or SEM (in capital letters, respectively). Please clarify, and correct accordingly throughout the manuscript.

Response: Thank you for pointing it out. We have corrected it to “SEM” throughout our revised manuscript.

3. Line 124, Figure 1: change y-axis to mg instead of g; indicate more clearly whether this is weight per lodicule, water content per lodicule?

Response: Thanks for the suggestion. Due to the tiny size of the lodicules, we measured the water content of 100 pairs lodicules at each time point in triplicate. Therefore, Figure 1i indicates the water content of 100 pairs lodicules. To make it clearly, we changed y-axis label to “Water content of 100 pairs lodicules”, and use mg as the unit in the revised version.

4. Line 156, Supplementary Fig. 2: Abbreviations used in the figure need a better explanation. What means MF and BP? Readers have to guess. Suggestion: make a legend where you explain all the abbreviations such a T9, TF etc., and use it in each figure legend where it is used. It will help the reader to get the point faster.

Response: Thank you for pointing it out. MF means molecular function. BF means biological process. As suggested, we have explained all abbreviations in the Figure Legends.

5. Line 180: Correct: write either ‘leaf’ ore ‘leaves’

Response: Thanks. We have corrected it.

6. Line 210, Fig. 4a: better label upper row as free-GFP, lower row as OsMYB8-GFP. The current labelling causes confusion.

Response: We apologize for this confusion. We have corrected it as suggested.

Reviewer #4 (Remarks to the Author):

The manuscript "Natural variation in OsMYB8 confers diurnal floret opening time divergence between indica and japonica subspecies" investigates the mechanism that regulates daily flower (floret) opening and closure in rice. This topic is of note as the natural variation of locus for transcription factor OsMYB8 showed potential of synchronous floret opening important for more efficient harvesting of hybrid rice seeds.

Through well-designed sampling and global gene expression analysis, the authors identified the transcription factor MYB8 as a responsive factor for synchrony. Lodicule is the mechanical organ driving the movement. In lodicules, MYB8 expression profiles are consistent with the occurrence of the movement. Molecular analyses to probe binding sites isolated the target JA-Ile promoter motifs. Genetic approach making these CRISPER/Cas9 induced mutant lines determined the role of inducer for the target genes. Further authors found that the indica MYB8 locus has dominant positive promoter activity and demonstrated the application in japonica to establish the strong synchrony.

Overall, this is an outstanding manuscript contributing to the research field. The experiments in this manuscript are well thought out and organized. Previously a DFOT factor, DUF642, is determined, which negatively regulates rice DFOT through lodicule cell-wall modulation, in this report authors demonstrated another genetic DFOT phenotype (variation) due to promoter haplotypes. The finding and application are outstanding noteworthy points in plant and the breeding sciences. However, there are a few details that would be added to the text to solve difficulties in finding the words and means for general readers in the journal.

Response: We appreciate the reviewer's careful review as well as positive and constructive comments on our manuscript.

Major notes:

Strong points: Consistency of the context. Seamless in experimental results from lodicule expression to functional analysis among Haplotype of promoter, MYB8 binding to JAR1 promoter and CRISPER/Cas9 gene-targeting mutation.

Response: Thanks very much for your support.

Weak points:

Fig. 5a shows 2 electropherograms as DNA sequencing of Crispr/cas9 derivatives. The Osjar1#2 sequence looks like 2 base pair deletion. But it might be one base pair deletion. It should be sequenced from the lower side with an anti-directional oligonucleotide primer.

Response: Thanks for your suggestion. As suggested, we have re-sequenced the *OsJAR1* gene in the *Osjar1#2* mutant lines using directional and anti-directional oligonucleotide primers. The results clearly showed that the *Osjar1#2* mutant indeed carries 1-bp deletion. We have added the new sequencing peak diagram in Figure 5a in our revised manuscript.

Fig. 6a shows haplotypes of OsMYB8 promoter. These OsMYB8 promoters determine the strength of transcription. Although authors determined 3 variations in the promoter, readers would expect other related regulatory motifs, e.g., for lodicule specific expression and light response. Please describe in discussion or supplementary figure.

Response: Thanks for the suggestion. We analyzed the *cis*-acting elements for transcriptional factors in the 2-kb *OsMYB8* promoter using PLANTCARE (<http://bioinformatics.psb.ugent.be/webtools/plantcare/html/>). We found that the *japonica* and *indica* promoters contained various response elements, such as light response, temperature response and plant hormone response. Although none of the six SNPs in the promoter between *japonica* and *indica* are located in these regulatory elements, it will be very interesting to figure out whether *OsMYB8* could integrate external factors with internal hormone signaling to regulate DFOT in future studies. We have discussed this point in the revised manuscript (line 469-476) as follows:

“It is also worth noting that DFOT is easily affected by environmental factors, such as light, temperature and humidity. We found that the *OsMYB8* promoters of *japonica* and *indica* varieties contain various *cis*-elements responsive to light, temperature and hormones based on sequence analysis with PLANTCARE (<http://bioinformatics.psb.ugent.be/webtools/plantcare/html/>). How these external factors influence the expression of *OsMYB8*, and thus DFOT will be an interesting avenue for future research.”

Minor notes:

Line 95 *OsMYB8* is italic now. Not Italic.

Response: Thanks. We have corrected it.

Line 159 The T9 time point were,, was?

Response: We apologize for the mistake. It should be “was”, and we have corrected it!

Line 239 log2FC, log2FC (folds change).

Response: Thanks. We have changed it.

Line 265 to regulates, regulate.

Response: Thanks. We have corrected it

Line 298 Petin is Pectin.

Response: We apologize for the typing error. We have corrected it.

Line 317 (TrJ) is italic? What name is referred to Aus?

Response: Thanks. We have corrected it. “*Aus*” refers to a subpopulation of rice known as *Aus* rice or *Aus* paddy rice. It is considered to be originated predominantly in stress-prone regions, such as Bangladesh and India³⁴. *Aus* rice exhibits rich genetic diversity and is a promising source for the development of new stress-tolerant rice cultivars³⁵.

Line 580 What is PLB?

Response: Thanks. PLB stand for passive lysis buffer, we have clarified it.

Line 727 32. Tsutomu, I. et al. ,, Ishimaru, T. et al.

Response: Thanks. We have corrected it.

Supplementary Fig. 9 Line 114 indicate is indicates.

Response: Thanks. We have corrected it.

References

1. He, Y., Lin, Y. & Zeng, X. Dynamic changes of jasmonic acid biosynthesis in rice florets during natural anthesis. *Acta Agron. Sin.* **38**, 1891-1899 (2012).
2. Li, S. et al. Synthetic biosensor for mapping dynamic responses and spatio-temporal distribution of jasmonate in rice. *Plant Biotechnol. J.* **19**, 2392-2394 (2021).
3. Ye, H., Du, H., Tang, N., Li, X. & Xiong, L. Identification and expression profiling analysis of TIFY family genes involved in stress and phytohormone responses in rice. *Plant Mol. Biol.* **71**, 291-305 (2009).
4. Chico, J. M., Chini, A., Fonseca, S. & Solano, R. JAZ repressors set the rhythm in jasmonate signaling. *Curr. Opin. Plant Biol.* **11**, 486-494 (2008).
5. Chini, A. et al. The JAZ family of repressors is the missing link in jasmonate signalling. *Nature* **448**, 666-671 (2007).
6. Fu, Y., Xiang, M., Jiang, H., He, Y. & Zeng, X. Transcriptome profiling of lodicules before floret opening in *Oryza sativa* L. *Sci. Agric. Sin.* **49**, 1017-1033 (2016).
7. Yan, Z., Deng, R., Zhang, H., Li, J. & Zhu, S. Transcriptome analysis of floret opening and closure both *Indica* and *Japonica* rice. *3 Biotech* **12**, 188 (2022).
8. Yan, Z., Xu, H., Ma, Z., Gao, D. & Xu, Z. Differential response of floret opening to exo-methyl jasmonate between subsp. *indica* and subsp. *japonica* in rice. *Sci. Agric. Sin.* **47**, 2529-2540 (2014).
9. Wang, Z., Gu, Y. & Gao, Y. Studies on the mechanism of the anthesis of rice III. structure of the lodicule and changes of its contents during flowering. *The Crop J.* **17**, 96-101 (1991).
10. Heslop-Harrison, Y. & Heslop-Harrison, J. S. Lodicule function and filament extension in the grasses: potassium ion movement and tissue specialization. *Ann. Bot.* **77**, 573-582 (1996).
11. Qin, Y., Yang, J. & Zhao, J. Calcium changes and the response to methyl jasmonate in rice lodicules during anthesis. *Protoplasma* **225**, 103-112 (2005).
12. Damaris, R., Lin, Z., Yang, P. & He, D. The rice alpha-amylase, conserved regulator of seed maturation and germination. *Int. J. Mol. Sci.* **20** 450 (2019).
13. Opassiri, R. et al. Analysis of rice glycosyl hydrolase family 1 and expression of

- Os4bglu12* β -glucosidase. *BMC Plant Biol.* **6**, 33 (2006).
14. Hu, Z. et al. Rice SUT and SWEET transporters. *Int. J. Mol. Sci.* **22**, 11198 (2021).
 15. Sampedro, J. & Cosgrove, D. J. The expansin superfamily. *Genome Biol.* **6**, 242 (2005).
 16. Yokoyama, R., Rose, J. K. C. & Nishitani, K. A surprising diversity and abundance of xyloglucan endotransglucosylase/hydrolases in rice. Classification and expression analysis. *Plant Physiol.* **134**, 1088-1099 (2004).
 17. Liang, Y. et al. A comparative analysis of the evolution, expression, and cis-regulatory element of polygalacturonase genes in grasses and dicots. *Funct. Integr. Genomics* **16**, 641-656 (2016).
 18. Sakurai, J., Ishikawa, F., Yamaguchi, T., Uemura, M. & Maeshima, M. Identification of 33 rice aquaporin genes and analysis of their expression and function. *Plant & cell physiol.* **46**, 1568-1577 (2005).
 19. Ghorbel, M., Brini, F., Sharma, A. & Landi, M. Role of jasmonic acid in plants: the molecular point of view. *Plant Cell Rep.* **40**, 1471-1494 (2021).
 20. Huang, H., Chen, Y., Wang, S., Qi, T. & Song, S. Jasmonate action and crosstalk in flower development and fertility. *J. Exp. Bot.* **74**, 1186-1197 (2023).
 21. Chopy, M. et al. A single MYB transcription factor with multiple functions during flower development. *New Phytol.* **239**, 2007-2025 (2023).
 22. Wang, Y. et al. MYB transcription factors and their roles in the male reproductive development of flowering plants. *Plant Sci.* **335**, 111811 (2023).
 23. Reeves, P. H. et al. A regulatory network for coordinated flower maturation. *PLoS Genet.* **8** e1002506 (2012).
 24. Huang, H., Liu, B., Liu, L. & Song, S. Jasmonate action in plant growth and development. *J. Exp. Bot.* **68**, 1349-1359 (2017).
 25. Schubert, R. et al. Tomato MYB21 acts in ovules to mediate jasmonate-regulated fertility. *The Plant Cell* **31**, 1043-1062 (2019).
 26. Riemann, M., Riemann, M. & Takano, M. Rice *JASMONATE RESISTANT 1* is involved in phytochrome and jasmonate signalling. *Plant Cell & Environ.* **31**, 783-792 (2010).

27. Riemann, M., Haga, K., Shimizu, T., Okada, K. & Iino, M. Identification of rice *ALLENE OXIDE CYCLASE* mutants and the function of jasmonate for defence against *Magnaporthe oryzae*. *The Plant J.* **74**, 226-238 (2013).
28. Inagaki, H. et al. Genome editing reveals both the crucial role of OsCOI2 in jasmonate signaling and the functional diversity of COI1 homologs in rice. *Plant Cell Physiol.* **64**, 405-421 (2023).
29. Svyatyna, K. et al. Light induces jasmonate-isoleucine conjugation via *OsJARI*-dependent and -independent pathways in rice. *Plant Cell Environ.* **37**, 827-839 (2014).
30. Yon, F. et al. Silencing *Nicotiana attenuata LHY* and *ZTL* alters circadian rhythms in flowers. *New Phytol.* **209**, 1058-1066 (2016).
31. Ge, J. et al. Genome-wide selection and introgression of Chinese rice varieties during breeding. *J. Genet. Genomics* **49**, 492-501 (2022).
32. Wang, M., Chen, M., Huang, Z., Zhou, H. & Liu, Z. Advances on the study of diurnal flower-opening times of rice. *Int. J. Mol. Sci.* **24**, 10654 (2023).
33. Xiao, Y. et al. *OsJARI* is required for JA-regulated floret opening and anther dehiscence in rice. *Plant Mol. Biol.* **86**, 19-33 (2014).
34. Chen, C., Norton, G. J. & Price, A. H. Genome-wide association mapping for salt tolerance of rice seedlings grown in hydroponic and soil systems using the bengal and assam *Aus* panel. *Front. Plant Sci.* **11** 576479 (2020).
35. Liao, Q. et al. *Aus* rice root architecture variation contributing to grain yield under drought suggests a key role of nodal root diameter class. *Plant Cell Environ.* **45**, 854-870 (2022).

Reviewers' Comments:

Reviewer #1:

Remarks to the Author:

In the revised manuscript, the authors have addressed the feedback provided by the reviewers, and their responses are deemed satisfactory. However, additional concerns regarding the data supporting their conclusions have surfaced.

Regarding Response Figure 1, the source of the fluorescence signal remains ambiguous. It is imperative to ascertain if all figures emanate from the same lodicule position. Moreover, the incorporation of a negative control from mJ6V lines is indispensable to establish the correlation between VENUS signal alterations and lodicule development.

Within supplementary Figure 7, samples in panels a and c correspond to stages preceding flower opening. Nevertheless, the conspicuous difference in coloration between these panels requires elucidation.

Could the authors furnish information on the seed-setting rate in the *Osmyb8* mutant?

A critical query arises concerning the authenticity of the assertion that the *OsMYB8TF* haplotype in the ZH11 background genuinely augments inter-subspecific indica-japonica hybrid seed production. To substantiate this claim, it would be advantageous for the authors to present supporting data.

The assertion in the abstract and introduction sections, purporting that *OsMYB8* regulates the expression of genes associated with cell osmolality and cell wall remodeling in lodicules to facilitate floret opening, lacks sufficient cellular and genetic evidence. Notably, the observation that florets of the *Osmyb8* CRISPR mutant still open at 12:00 noon, coupled with the independence of JA-induced *OsJAR1* expression from *OsMYB8*, suggests that *OsMYB8* is not the exclusive regulator governing JA biosynthesis and lodicule swelling during rice floret opening.

Beyond the inherent variations in the promoter sequence of *OsMYB8*, it is imperative to discuss the role of the spatial-temporal expression pattern of *OsMYB8*, particularly in light of the changes in DFOT across different seasons.

Reviewer #2:

Remarks to the Author:

The authors have done several additional experiments in response to previous reviews, and the manuscript is about ready to publish. I have just one further comment, related to their response to one of my previous comments. They have looked at expression of the MYB8-like gene, and present a figure in their response to reviews. As they note, this gene has high expression in anthers, but it is also expressed in lodicules. Thus, it seems likely to me that this MYB8-like gene functions in both anthers and lodicules. For example, a *myb8 myb8-like* double mutant might have a stronger lodicule opening defect than does the *myb8* single mutant. While I agree that to characterize function of the MYB8-like gene would be beyond the scope of the current work, I suggest to say something about these ideas. For example they could show their MYB8-like gene expression data in Figure S4; and in the Discussion (perhaps after line 442) they could discuss the possibly redundancy, as well as the possibility that MYB8-like is the main JA pathway regulator in anthers - that is, this MYB subfamily may perhaps have a conserved role in monocots and dicots.

Reviewer #4:

Remarks to the Author:

I was pleased to be able to read the revised version of this manuscript. It is very readable with some additional experiments and corrections. I look forward to future developments, such as which site of the promoter determines the specificity of the lodicule.

The paper became more persuasive by adding additional experiments to the reviewers' points and requests. Also, sufficient additions have been made to consider the requirements. Therefore, readers' questions are answered each time, making it very easy to read. In addition, concerns raised by another reviewer regarding the gene/genotype labeling of transformants have been solved by unifying the labeling. Therefore, it is expected that this submitted paper will be accepted promptly.

Below are minor suggestion and a few typos.

Line 654 Although the authors seem to quantify DNA fragment of the OsMYC8 by using real time qPCR (RT-qPCR), I easily read it Reverse Transcriptional qPCR. Such that, I recommend the RT should be 'real time'.

Line 579-580 50-100ng, 250nM, 250nM need spaces between number and unit.

Responses to reviewers' comments

Reviewers' comments:

Reviewer #1 (Remarks to the Author):

In the revised manuscript, the authors have addressed the feedback provided by the reviewers, and their responses are deemed satisfactory. However, additional concerns regarding the data supporting their conclusions have surfaced.

Response: We appreciate the reviewer's positive comments and valuable suggestions.

Regarding Response Figure 1, the source of the fluorescence signal remains ambiguous. It is imperative to ascertain if all figures emanate from the same lodicule position. Moreover, the incorporation of a negative control from mJ6V lines is indispensable to establish the correlation between VENUS signal alterations and lodicule development.

Response: Thanks for the reviewer's preciseness. In our study, we utilized the *Jas-VENUS*, a fully validated biosensor for *in vivo* visualization of JA perception in multiple previous studies^{1,2}. The J6V and mJ6V vectors used in our study were previously constructed and functionally confirmed by Li *et al*². The cells selected to capture fluorescence signals were from the nearly identical positions where the lodicules enlarge dramatically, as indicated by yellow boxes in Response Fig. 1a. As suggested, we further analyzed fluorescent signals observed in lodicules of the mJ6V plants, and captured similar fluorescence intensities at different time points (Response Fig. 2). This result confirmed that the J6V plants analyzed in this study indeed specifically responded to JA.

Response Fig. 1 VENUS fluorescence of the J6V transgenic plant lodicule cells at different time points. a Lodicule morphology of the J6V transgenic plant at 18:00 on the day before floret opening and different time points on the day of floret opening, including 8:00 am, 9:00 am, 10:00 am, 11:00 am and 12:00 noon (peak floret opening time for ZH11) in October 2023 in Guangzhou. The yellow boxes indicate the position observed under the super-resolution laser scanning confocal microscope. Scale bars, 250 μm . **b** VENUS signals of the J6V transgenic plant lodicule cells at different time points. Scale bars, 10 μm . **c** Quantification of VENUS fluorescence normalized to mCherry signals at different time points. Values are mean \pm SEM. (n=6). **d** Lodicule volume of ZH11 at different time points. Values are mean \pm SEM. (n=10).

Response Fig. 2 VENUS fluorescence of the mJ6V transgenic plant lodicule cells at different time points. **a** Lodicule morphology of the mJ6V transgenic plant at 18:00 on the day before floret opening and different time points on the day of floret opening, including 8:00 am, 9:00 am, 10:00 am, 11:00 am and 12:00 noon (peak floret opening time for ZH11) in December 2023 in Guangzhou. The yellow boxes indicate the position observed under the super-resolution laser scanning confocal microscope. Scale bars, 250 μ m. **b** VENUS signals in the mJ6V transgenic plant lodicule cells at different time points. Scale bars, 10 μ m. **c** Quantification of VENUS fluorescence normalized to mCherry signals at different time points. Values are mean \pm SEM. (n = 3).

Within supplementary Figure 7, samples in panels a and c correspond to stages preceding flower opening. Nevertheless, the conspicuous difference in coloration between these panels requires elucidation.

Response: Thanks for the reminding. In supplementary Fig. 7a, the image exhibits the florets predominantly in green color, while supplementary Fig. 7c focused on the stamens,

which are yellow color. To ensure clear pictures capture, they were taken under different white balance conditions, thus leading to the conspicuous different colorations between these figures.

Could the authors furnish information on the seed-setting rate in the *Osmyb8* mutant?

Response: We appreciate the reviewer's valuable suggestions. As suggested, we analyzed the seed setting rate in the *Osmyb8* mutant. The setting rates of ZH11, *Osmyb8^{ZH}#1* and *Osmyb8^{ZH}#2* are 94.85%, 95.67% and 93.93%, respectively, exhibiting a comparable seed setting rate; while the setting rates of TFB, *Osmyb8^{TF}#1* and *Osmyb8^{TF}#2* are 90.27%, 85.86% and 88.60%, respectively, with a slightly lower seed setting rate in *Osmyb8^{TF}* compared to TFB (New Supplementary Fig. 7f). These results further confirmed that *OsMYB8* does not significantly influence panicle fertility in rice.

New Supplementary Fig. 7 *OsMYB8* did not affect rice fertility and anther dehiscence.

a, b Florets of ZH11, *Osmyb8^{ZH}*, TFB and *Osmyb8^{TF}* before opening (**a**) and after opening

(b). Scale bars, 1 mm. **c** Stamens of ZH11, *Osmyb8^{ZH}*, TFB and *Osmyb8^{TF}* before opening. Scale bars, 1 mm. **d** Images of pollens of ZH11, *Osmyb8^{ZH}*, TFB and *Osmyb8^{TF}* stained with 1% I₂-KI (potassium iodide) solution. Scale bars, 50 μm. **e** Pollen fertility of ZH11 and *Osmyb8^{ZH}*, TFB and *Osmyb8^{TF}* based on I₂-KI staining. Values are means ± SEM. (n = 10). Significance is evaluated by the two-sided Student's *t*-test. **f** Seed setting rates of ZH11 and *Osmyb8^{ZH}*, TFB and *Osmyb8^{TF}*. Values are means ± SEM. (n = 10). Significance is evaluated by the two-sided Student's *t*-test.

A critical query arises concerning the authenticity of the assertion that the *Osmyb8^{TF}* haplotype in the ZH11 background genuinely augments inter-subspecific indica-japonica hybrid seed production. To substantiate this claim, it would be advantageous for the authors to present supporting data.

Response: We appreciate the reviewer's comments. During commercial production of hybrid seeds, plants of the female parent (usually a male sterile line) are planted in rows side-by-side with plants of the male parent, and pollination is aided by humans or mechanicals (such as unmanned-helicopters). Successful hybrid seed production requires the synchronization of diurnal floret opening time (DFOT) between the parental lines. DFOT is categorized into three stages: initial DFOT (the time when the first floret opens), peak DFOT (the time when the number of opened florets reaches 50% of the anticipated florets for the day), and terminal DFOT (the time when all florets closed). According to the study examining the DFOT of 25 *indica* varieties and 47 *japonica* varieties at Jiaxing (30°21'N, 121°16'E) in August 2003 and at LingShui (18°22'N, 110°08'E) in March 2004 (Response Table 1), the *indica* and *japonica* varieties exhibited a DFOT of 8:20-9:30 am and 10:10-12:55 pm in Jiaxing, respectively, showing no overlap in DFOT between the *indica* and *japonica* varieties. Similarly, in LingShui, *indica* varieties exhibited a DFOT from 10:05-11:45 am (with a peak DFOT between 10:20-11:10 am), while the florets of *japonica* varieties opened from 11:00-13:30 pm (with a peak DFOT between 11:30-12:55 pm), with no overlap in peak DFOT between the two subspecies³. This asynchronous DFOT between *indica* and *japonica* parental lines caused a low hybrid seed yield (~50 kg/mu), which is far below the minimum yield (300 kg/mu) for commercial demand.

Breeding *japonica* with earlier DFOT could effectively elevate the yield of hybrid seed production. For instance, a *japonica* male sterile line Zhe08A exhibited an ~20 min earlier DFOT than its primitive parental line Zhe04A through genetic improvement, and the seed production yield achieving up to 120 kg/mu when crossed with *indica* restorer line Zhehui F1015⁴.

In our study, the *OsMYB8^{TF}* haplotype could promote DFOT by 0.5-1 h in ZH11 (Fig. 2k), enabling it to significantly narrow the DFOT interval and greatly expand DFOT overlap of the *indica* and *japonica* (Response Fig. 3). Therefore, with the expanded DFOT overlap, hybrid seed yield could be greatly enhanced by mechanical assistance during commercial *indica-japonica* hybrid seed production. However, we are unable to directly test this using our materials, as ZH11 and *OsMYB8^{TF}/ZH11* are fertile and self-pollinating plants (hybrid seed production uses sterile maternal lines to cross with the paternal line). Generating the sterile and maintainer lines carrying *OsMYB8^{TF}* transgene will take at least two years, and we think the evidence for supporting our claim will be incremental, but not essential.

Table 1 The DFOT of *indica* and *japonica*³

Place	Jiaying			LingShui		
	Initial DFOT	Peak DFOT	Terminal DFOT	Initial DFOT	Peak DFOT	Terminal DFOT
indica	8:20-9:05	8:50-9:30	9:10-10:00	10:05-10:35	10:20-11:10	10:55-11:45
japonica	10:10-11:30	10:35-12:15	11:05-12:55	11:00-12:15	11:30-12:55	11:50-13:30

Response Fig. 3 A diagram of *indica* and *japonica* DFOT. a, b The hypothetical diagram of *indica* and *japonica* DFOT at JiaXing in August 2003 (a) at LingShui in March 2004 (b). Solid red and blue lines indicated DFOT of *indica* and *japonica* varieties, and blue dotted lines indicated conjectural DFOT of *japonica* varieties carrying *OsMYB8^{Ind}* haplotype. The DFOT data are obtained from Li *et al*³.

The assertion in the abstract and introduction sections, purporting that *OsMYB8* regulates the expression of genes associated with cell osmolality and cell wall remodeling in lodicules to facilitate floret opening, lacks sufficient cellular and genetic evidence. Notably, the observation that florets of the *Osmyb8* CRISPR mutant still open at 12:00 noon, coupled with the independence of JA-induced *OsJAR1* expression from *OsMYB8*, suggests that *OsMYB8* is not the exclusive regulator governing JA biosynthesis and lodicule swelling during rice floret opening.

Response: We appreciate the reviewer's comments. Multiple cytological and physiological analyses have provided ample evidence to demonstrate that the floret opening is attributed to the alterations of cell turgor and osmotic pressure in lodicules⁵⁻¹⁰. Consistently, a series of transcriptome analysis of lodicules during anthesis revealed dynamic expression changes

in genes related to cellular osmotic regulation, such as carbohydrate metabolism, sugar transporters and aquaporins; and cell wall remodeling, such as polygalacturonase, pectin methyltransferase and cell wall-loosening proteins^{11,12}. Additionally, the *Diurnal Flower Opening Time 1 (DFOT1) / Early Morning Flowering1 (EMF1)* negatively regulates rice DFOT through modulating pectin methylesterification of lodicule cell-wall^{13,14}. The crucial role of JA in floret opening and closure regulation in grasses is also well documented¹⁵⁻¹⁸. Notably, several studies further confirmed that JA can induce soluble sugar and potassium ion accumulation in lodicules, thus promoting floret opening¹⁹⁻²². Moreover, mutations in JA biosynthesis gene *OsOPR7* and *OsJAR1* caused seriously disrupted floret opening and closure phenotype^{23,24}. Expression analyses showed that genes responsible for sucrose transport (*OsSWEET4/5/11/14/15*) and K⁺ homeostasis (*OsCHX14*), were down-regulated in lodicules of *Osopr7* and *Osjar1* respectively^{24,25}. These studies indicate that JA could promote osmotic substance accumulation in lodicules and lodicule expansion.

In our study, we found altered expression of 10 genes associated with cell osmolality and cell wall remodeling in the *Osmyb8* and *Osjar1* mutants (Supplementary Fig. 11). In this revised manuscript, we further analyzed the soluble sugar content in the lodicules of ZH11 and *Osmyb8^{ZH}* collected at 10:00 am, and the result showed that the levels of sucrose, fructose and total soluble sugars were decreased in the lodicules of *Osmyb8^{ZH}* (New Supplementary Fig. 12). Therefore, we propose that *OsMYB8-OsJAR1* module could affect floret opening, probably by influencing the expression of genes related to cell osmolality and cell wall remodeling in lodicules.

In our study, we have not claimed that *OsMYB8* is the exclusive regulator governing JA biosynthesis and lodicule swelling during rice floret opening, we just demonstrated *OsMYB8* as an important gene for DFOT divergence through regulating variations in JA-Ile levels between *indica* and *japonica*. Considering that DFOT is also regulated by various endogenous plant hormones such as auxin, ABA and ethylene, and is easily affected by external environmental factors such as light, temperature and humidity, it is sure that there would be other regulators involved in governing JA biosynthesis and lodicule swelling during rice floret opening.

New Supplementary Fig. 12 Determination of soluble sugar content in the lodicules of ZH11 and *Osmyb8^{ZH}*. The sucrose, glucose, fructose and total sugar contents in the lodicules of ZH11 and *Osmyb8^{ZH}* at 10:00 am. Values are mean \pm SEM. (n = 3 biological replicates). Significance is evaluated by the two-sided Student's *t*-test, and *P* values are indicated.

Beyond the inherent variations in the promoter sequence of OsMYB8, it is imperative to discuss the role of the spatial-temporal expression pattern of OsMYB8, particularly in light of the changes in DFOT across different seasons.

Response: Thank you for your suggestion. In this study, we showed that both ZH11 and TFB exhibited an earlier DFOT in the early season (June) than that in late season (October) (Fig. 1d,e), which may be due to the higher temperature and longer light period in June than that in October. We also found that the expression levels of *OsMYB8* were gradually increased towards opening time, reaching a peak expression level at 1 h before floret opening in both ZH11 and TFB (Supplementary Fig. 5). Considering the identification of various cis-elements responsive to light and temperature in the *OsMYB8* promoters, we speculated that the expression of *OsMYB8* might be influenced by light and temperature, thus affecting the DFOT of rice in different breeding seasons. We have discussed this point in lines 478-483.

Reviewer #2 (Remarks to the Author):

The authors have done several additional experiments in response to previous reviews, and the manuscript is about ready to publish. I have just one further comment, related to their response to one of my previous comments. They have looked at expression of the MYB8-like gene, and present a figure in their response to reviews. As they note, this gene has high expression in anthers, but it is also expressed in lodicules. Thus, it seems likely to me that this MYB8-like gene functions in both anthers and lodicules. For example, a *myb8 myb8*-like double mutant might have a stronger lodicule opening defect than does the *myb8* single mutant. While I agree that to characterize function of the MYB8-like gene would be beyond the scope of the current work, I suggest to say something about these ideas. For example they could show their MYB8-like gene expression data in Figure S4; and in the Discussion (perhaps after line 442) they could discuss the possibly redundancy, as well as the possibility that MYB8-like is the main JA pathway regulator in anthers - that is, this MYB subfamily may perhaps have a conserved role in monocots and dicots.

Response: We appreciate the reviewer's insightful comments. As suggested, we further analyzed the expression pattern of *OsMYB8-like*, and the result was shown as New Supplementary Fig. 16. Notably, the expression of *OsMYB8-like* gene was higher in anther compared to other floret organs, implying a probable function in the anther development regulation. We discussed this point in lines 453-458.

Supplementary Fig. 16 Expression pattern analysis of *OsMYB8-like*. **a** Protein sequence comparison of *OsMYB8* and *OsMYB8-like*. **b** Relative expression levels of *OsMYB8-like* in various tissues of ZH11. Values are mean ± SEM. (n = 3 biological replicates).

Reviewer #4 (Remarks to the Author):

I was pleased to be able to read the revised version of this manuscript. It is very readable with some additional experiments and corrections. I look forward to future developments, such as which site of the promoter determines the specificity of the lodicule.

Response: Thank you for your constructive comments, this would be an interesting avenue for our future research.

The paper became more persuasive by adding additional experiments to the reviewers' points and requests. Also, Sufficient additions have been made to consider the requirements. Therefore, readers' questions are answered each time, making it very easy to read. In addition, concerns raised by another reviewer regarding the gene/genotype labeling of transformants have been solved by unifying the labeling. Therefore, it is expected that this submitted paper will be accepted promptly.

Response: We are grateful of the reviewer's insightful comments.

Below are minor suggestion and a few typos.

Line 654 Although the authors seem to quantify DNA fragment of the OsMYB8 by using real time qPCR (RT-qPCR), I easily read it Reverse Transcriptional qPCR. Such that, I recommend the RT should be 'real time'.

Response: Thank you for pointing it out. We have corrected it in our revised manuscript.

Line 579-580 50-100ng, 250nM, 250nM need spaces between number and unit.

Response: Thank you for pointing it out. We have corrected it in our revised manuscript.

References

1. Larrieu, A. et al. A fluorescent hormone biosensor reveals the dynamics of jasmonate signalling in plants. *Nat. Commun.* **6**, 6043 (2015).
2. Li, S. et al. Synthetic biosensor for mapping dynamic responses and spatio-temporal distribution of jasmonate in rice. *Plant Biotechnol. J.* **19**, 2392-2394 (2021).
3. Li, J., Fan, G., Zhang, R. & Gao, R. Comparative experiment of flower opening time in different rice cultivars. *J. Zhejiang Agric. Sci.* **1**, 63-66 (2007).
4. Zhang, L. et al. Breeding and application of *japonica* rice sterile line Zhe08A. *J. Zhejiang Agric. Sci.* **58**, 1120-1122 (2017).
5. Wang, Z., Gu, Y. & Gao, Y. Studies on the mechanism of the anthesis of rice III. structure of the lodicule and changes of its contents during flowering. *Acta Agron. Sin.* **17**, 96-101 (1991).
6. Heslop-Harrison, Y. & Heslop-Harrison, J. S. Lodicule function and filament extension in the grasses: potassium ion movement and tissue specialization. *Ann. Bot.* **77**, 573-582 (1996).
7. Zeng, X., Zhou, X. & Wu, X. Advances in study of opening mechanism in rice florets. *Sci. Agri. Sin.* **37**, 188-195 (2004).
8. van Doorn, W. G. Flower opening and closure: a review. *J. Exp. Bot.* **54**, 1801-1812 (2003).

9. Gookin, T. E., Hunter, D. A. & Reid, M. S. Temporal analysis of alpha and beta-expansin expression during floral opening and senescence. *Plant Sci.* **164**, 769-781 (2003).
10. Zhao, Z. et al. Auxin regulates source-sink carbohydrate partitioning and reproductive organ development in rice. *Proc. Natl Acad. Sci. USA* **119** (2022).
11. Fu, Y., Xiang, M., Jiang, H., He, Y. & Zeng, X. Transcriptome profiling of lodicules before floret opening in *Oryza sativa* L. *Sci. Agric. Sin.* **49**, 1017-1033 (2016).
12. Yan, Z., Deng, R., Zhang, H., Li, J. & Zhu, S. Transcriptome analysis of floret opening and closure both *Indica* and *Japonica* rice. *3 Biotech.* **12**, 188 (2022).
13. Wang, M. et al. Methylesterification of cell-wall pectin controls the diurnal flower-opening times in rice. *Mol. Plant* **15**, 956-972 (2022).
14. Xu, P. et al. *EARLY MORNING FLOWERING1 (EMF1)* regulates the floret opening time by mediating lodicule cell wall formation in rice. *Plant Biotechnol. J.* **20**, 1441-1443 (2022).
15. Zeng, X. et al. Opening of rice floret in rapid response to methyl jasmonate. *J. Plant Growth Regul.* **18**, 153-158 (1999).
16. Gao, X., Zeng, X., Wang, S., Lin, P. & Zhou, X. Effect of methyl jasmonate and salicylic acid on the blossom of spikelets in sorghum and sudenese grass. *Chin. Agric. Sci. Bull.* **16**, 7-9 (2003).
17. Song, P. et al. Differential response of floret opening in male-sterile and male-fertile rices to methyl jasmonate. *Acta Botanica Sinica* **43**, 480-485 (2005).
18. Yan, Z., Xu, H., Ma, Z., Gao, D. & Xu, Z. Differential response of floret opening to exo-methyl jasmonate between subsp. *indica* and subsp. *japonica* in rice. *Sci. Agric. Sin.* **47**, 2529-2540 (2014).
19. Zou, C., He, Y. & Zeng, X. Study on mechanism of MeJA induced spikelets opening in rice. *2007 National Symposium on Plant Growth Substances* (2007).
20. Liu, L. et al. Jasmonic acid deficiency leads to scattered floret opening time in cytoplasmic male sterile rice Zhenshan 97A. *J. Exp. Bot.* **68**, 4613-4625 (2017).
21. Chen, J. et al. Physiological mechanism underlying the effect of high temperature during anthesis on spikelet-opening of photo-thermo-sensitive genic male sterile rice

- lines. *Sci. Rep.* **10**, 2210 (2020).
22. Yang, J. et al. Jasmonates alleviate spikelet-opening impairment caused by high temperature stress during anthesis of photo-thermo-sensitive genic male sterile rice lines. *Food Energy Secur.* **9** (2020).
23. Xiao, Y. et al. *OsJAR1* is required for JA-regulated floret opening and anther dehiscence in rice. *Plant Mol. Biol.* **86**, 19-33 (2014).
24. Li, X. et al. OPEN GLUME1: a key enzyme reducing the precursor of JA, participates in carbohydrate transport of lodicules during anthesis in rice. *Plant Cell Rep.* **37**, 329-346 (2018).
25. Chen, Y. et al. *OsCHX14* is involved in the K^+ homeostasis in rice (*Oryza sativa*) flowers. *Plant Cell Physiol.* **57**, 1530-1543 (2016).

Reviewers' Comments:

Reviewer #1:

Remarks to the Author:

The revised manuscript features enhanced clarity with the inclusion of more compelling evidence supporting the presented results. I have no further inquiries concerning this study. However, concerning the dynamic distribution of jasmonic acid (JA) during rice floret opening, it is recommended that Response Figures 1 and 2 be repositioned as Figure 5 or included in the supplementary figures. This adjustment is suggested as it marks the initial presentation of the dynamic changes in JA content within lodicule cells at distinct time points during floret opening.

Reviewer #2:

Remarks to the Author:

The authors have addressed my last remaining comment from the previous round of review.

Responses to reviewers' comments

Reviewers' comments:

Reviewer #1 (Remarks to the Author):

The revised manuscript features enhanced clarity with the inclusion of more compelling evidence supporting the presented results. I have no further inquiries concerning this study. However, concerning the dynamic distribution of jasmonic acid (JA) during rice floret opening, it is recommended that Response Figures 1 and 2 be repositioned as Figure 5 or included in the supplementary figures. This adjustment is suggested as it marks the initial presentation of the dynamic changes in JA content within lodicule cells at distinct time points during floret opening.

Response: We appreciate the reviewer's positive comments and valuable suggestions. Our study focused on the differential accumulation of jasmonic acid (JA) in the lodicules of *indica* and *japonica* subspecies, specifically at the time point before floret opening, which lead to differential diurnal floret opening time (DFOT) between the two subspecies. However, the dynamic changes in JA content within lodicule cells at distinct time points during floret opening we included in the last response letter is our preliminary result and awaits further substantiation, therefore, we prefer it should not be included in the final text at this stage.

Reviewer #2 (Remarks to the Author):

The authors have addressed my last remaining comment from the previous round of review.

Response: We are grateful of the reviewer's insightful comments.